# Molecular Geometry Pretraining with SE(3)-Invariant Denoising Distance Matching

**Shengchao Liu**
Mila - Québec AI Institute
Université de Montréal
*liusheng@mila.quebec*

**Hongyu Guo**
National Research Council Canada
University of Ottawa
*hongyu.guo@uottawa.ca*

**Jian Tang**
Mila - Québec AI Institute
HEC Montréal
CIFAR AI Chair
*jian.tang@hec.ca*

## Abstract

Molecular representation pretraining is critical in various applications for drug and material discovery due to the limited number of labeled molecules, and most existing work focuses on pretraining on 2D molecular graphs. However, the power of pretraining on 3D geometric structures has been less explored. This is owing to the difficulty of finding a sufficient proxy task that can empower the pretraining to effectively extract essential features from the geometric structures. Motivated by the dynamic nature of 3D molecules, where the continuous motion of a molecule in the 3D Euclidean space forms a smooth potential energy surface, we propose GeoSSL, a 3D coordinate denoising pretraining framework to model such an energy landscape. Further by leveraging an SE(3)-invariant score matching method, we propose GeoSSL-DDM in which the coordinate denoising proxy task is effectively boiled down to denoising the pairwise atomic distances in a molecule. Our comprehensive experiments confirm the effectiveness and robustness of our proposed method.

## 1 Introduction

Learning effective molecular representations is critical in a variety of tasks in drug and material discovery, such as molecular property prediction [14, 20, 21, 74], *de novo* molecular design and optimization [7, 36, 37, 39, 53, 77], and retrosynthesis and reaction planning [4, 22, 52, 64]. Recent work based on graph neural networks (GNNs) [20] has shown superior performance thanks to the simplicity and effectiveness of GNNs in modeling graph-structured data. However, the problem remains challenging due to the limited number of labeled molecules as it is in general expensive and time-consuming to label molecules, which usually requires expensive physics simulations or wet-lab experiments.

As a result, recently, there has been growing interest in developing pretraining or self-supervised learning methods for learning molecular representations by leveraging the huge amount of unlabeled molecule data [28, 35, 63, 75]. These methods have shown superior performance on many tasks, especially when the number of labeled molecules is insufficient. However, one limitation of these approaches is that they represent molecules as topological graphs, and molecular representations are learned through pretraining 2D topological structures (*i.e.*, based on the covalent bonds). But intrinsically, for molecules, a more natural representation is based on their 3D geometric structures, which largely determine the corresponding physical and chemical properties. Indeed, recent works [20, 38] have empirically verified the importance of applying 3D geometric information for molecular property prediction tasks. Therefore, a more promising direction is to pretrain molecular representations based on their 3D geometric structures, which is the main focus of this paper.

The main challenge for molecule geometric pretraining arises from discovering an effective proxy task to empower the pretraining to extract essential features from the 3D geometric structures. Our proxy task is motivated by the following observations. Studies [48] have shown that molecules are not static but in a continuous motion in the 3D Euclidean space, forming a potential energy surface (PES). As shown in Figure 1, it is desirable to study the molecule in the local minima of the PES, called *conformer*. However, such stable state conformer often comes with different noises for the following reasons. First, the statistical and systematic errors in conformation estimation are unavoidable [11]. Second, it has been well-acknowledged that a conformer can have vibrations

around the local minima in PES. Such characteristics of the molecular geometry motivate us to attempt to denoise the molecular coordinates around the local minima, to mimic the computation errors and conformation vibration within the corresponding local region. The denoising goal is to learn molecular representations that are robust to such noises and effectively capture the energy surface around the local minima.

To achieve the aforementioned goal, we first introduce a general *geometric self-supervised learning* framework called GeoSSL. Based on this, we further propose an *SE(3)-invariant denoising distance matching* pretraining algorithm, GeoSSL-DDM. In a nutshell, to capture the smooth energy surface around the local minima, we aim to maximize the mutual information (MI) between a given *stable geometry* and its *perturbed version* (*i.e.*, $g_1$ and $g_2$ in Figure 1). In practice, it is difficult to directly maximize the mutual information between two random variables. Thus, we propose to maximize an equivalent lower bound of the above mutual information, which amounts to a pretraining framework on denoising a geometric structure, coined GeoSSL. Moreover, directly denoising such noisy coordinates remains challenging because one may need to effectively constrain the pairwise atomic distances while changing the atomic coordinates. To cope with this obstacle, we further leverage an SE(3)-invariant score matching method, GeoSSL-DDM, to successfully transform the coordinate denoising desire to the denoising of pairwise atomic distances, which then can be effectively computed. In other words, our pretraining proxy task, namely mutual information maximization, effectively boils down to achieving an intuitive learning objective: denoising a molecule's pairwise atomic distances. Using 22 downstream geometric molecular prediction tasks, we empirically verify that our method outperforms nine pretraining baselines.

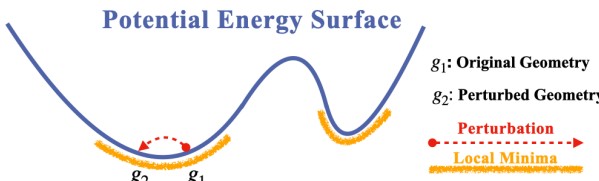

Figure 1: Illustration on coordinate geometry of molecules. The molecule is in a continuous motion, forming a potential energy surface (PES), where each 3D coordinate (x-axis) corresponds to an energy value (y-axis). The provided molecules, *i.e.*, conformers, are in the local minima ($g_1$). It often comes with noises around the minima (*e.g.*, statistical and systematic errors or vibrations), which can be captured using the perturbed geometry ($g_2$).

Our main contributions are summarized as follows. (1) We propose a novel geometric self-supervised learning framework, GeoSSL. To the best of our knowledge, it is the first pretraining framework focusing on the pure 3D molecular data [1]. (2) To overcome the challenge of attaining the coordinate denoising objective in GeoSSL, we propose GeoSSL-DDM, an SE(3)-invariant score matching strategy to successfully transform such objective into the denoising of pairwise atomic distances. (3) We empirically demonstrate the effectiveness and robustness of GeoSSL-DDM on 22 downstream tasks.

## 2 RELATED WORK

### 2.1 EQUIVARIANT GEOMETRIC MOLECULE REPRESENTATION LEARNING

**Geometric representation learning.** Recently, 3D geometric representation learning has been widely explored in the machine learning community, including but not limited to 3D point clouds [8, 44, 55, 67], N-body particle [45, 47], and 3D molecular conformation [6, 31, 32, 41, 50, 51, 56], amongst many others. The learned representation should satisfy the physical constraints, *e.g.*, it should be equivariant to the rotation and transition in the 3D Euclidean space. Such constraints can be described using group symmetry as introduced below.

**SE(3)-invariant energy.** Constrained by the physical nature of 3D geometric data, a key principle we need to follow is to learn an SE(3)-equivariant representation function. The SE(3) is the special Euclidean group consisting of rigid transformations in the 3D Cartesian space, where the transformations include all the combinations of translations and rotations. Namely, the learned representation should be equivariant to translations and rotations for molecule geometries. We also note that the representation function needlessly satisfies the reflection equivariance for certain tasks like molecular chirality [1]. For more rigorous discussion, please check [17, 19, 65]. In this work, we will design an SE(3)-invariant energy (score) function in addition to the SE(3)-equivariant representation backbone model.

---

[1]During the rebuttal of our submission, one of the reviewers pointed us to this parallel work [76], which is also under review. We provide a detailed comparison with this work in Appendix G.

## 2.2 SELF-SUPERVISED LEARNING FOR MOLECULE REPRESENTATION LEARNING

In general, there are two categories of self-supervised learning (SSL) [40, 42, 71, 72]: contrastive and generative, and the main difference is if the supervised signals are constructed in an inter-data or intra-data manner. Contrastive SSL extracts two views from the data and determines the supervised signals by detecting whether the sampled view pairs are from the same data. Generative SSL learns structural information by reconstructing partial information from the data itself.

**2D molecular graph (topology) self-supervised learning.** One of the mainstream research lines for molecule pretraining is on the 2D molecular graph. It treats the molecules as 2D graphs, where atoms and bonds are nodes and edges, respectively. It then carries out a pretraining task by either detecting if the two augmentations (*e.g.*, neighborhood extraction, node dropping, edge dropping, etc) correspond to the same molecular graph [28, 63, 75] or if the representation can successfully reconstruct certain substructures of the molecular graphs [28, 29, 35].

**3D molecular graph (geometry) self-supervised learning.** Self-supervised learning for 3D molecular graphs is still underexplored. The only related works are [16, 38], which leverage both 2D topology and 3D conformation to improve the molecule representation learning. For example, ChemRL-GEM [16] designs a novel model using 2D and 3D molecular graphs. Regarding SSL, it utilizes the geometry information by conducting distance and angle prediction as the generative pretraining tasks. GraphMVP [38] introduces an extra 2D topology and employs detection and reconstruction tasks simultaneously between 2D and 3D graphs, yet it focuses on 2D downstream tasks due to the small scale of the pretraining dataset. To the best of our knowledge, our work is the first to explicitly do SSL on pure 3D geometry along the molecule representation learning research line. We note that there is a parallel work [76], which is also under review; Appendix E provides a detailed comparison, highlighting the fact that the parallel work is a special case of GeoSSL-DDM.

## 3 PRELIMINARIES

**Molecular geometry graph.** Molecules can be naturally featured in a geometric formulation, *i.e.*, all the atoms are spatially located in 3D Euclidean space. Note that the covalent bonds are added heuristically by expert rules, so they are only applicable in 2D topology graphs. Besides, atoms are not static but in a continual motion along a potential energy surface [2]. The 3D structures at the local minima on this surface are named *conformer*, as shown in Figure 1. Conformers at such an equilibrium state possess nice properties, and we would like to model them during pretraining.

**Geometric neural network.** We denote each conformer as $g = (X, R)$. Here $X \in \mathbb{R}^{n \times d}$ is the atom attribute matrix and $R \in \mathbb{R}^{n \times 3}$ is the atom 3D-coordinate matrix, where $n$ is the number of atoms and $d$ is the feature dimension. The representations for the $i$-th node and whole molecule are:

$$h_i = \text{GNN-3D}(T(g))_i = \text{GNN-3D}(T(X, R))_i, \qquad h = \text{READOUT}(h_0, \ldots, h_{n-1}), \qquad (1)$$

where $T$ is the transformation function like atom masking, and READOUT is the readout function. In this work, we take the mean over all the node representations as the readout function.

**Energy-based model and denoising score matching.** Energy-based model (EBM) is a flexible and powerful tool for modeling data distribution. It has the form of Gibbs distribution as $p_\theta(\boldsymbol{x}) = \exp(-E(\boldsymbol{x}))/A$, where $p_\theta(\boldsymbol{x})$ is the model distribution and $A$ denotes the normalization constant. The computation of such probability is intractable due to the high cardinality of the data space. Recently, great progress has been made in solving this intractable function, including contrastive divergence [12], noise contrastive estimation [25], and score matching (SM) [30, 60, 61]. For example, SM solves this by first introducing the concept *score*, the gradient of the log-likelihood with respect to the data, and then matching the model score with the data score using Fisher divergence. This approach has been further improved by combining SM with denoising auto-encoding, forming the promising denoising score matching (DSM) strategy [69]. In this work, we will explore the potential of leveraging DSM for molecule geometry representation learning. We aim to utilize pairwise distance information, one of the most fundamental factors in the geometric molecule data.

**Problem setup.** Our goal here is to apply a self-supervised pretraining algorithm on a large molecular geometric dataset and adapt the pretrained representation for fine-tuning on geometric downstream tasks. For both the pretraining and downstream tasks, only the 3D geometric information is available, and our solution is agnostic in terms of the backbone geometric neural network.

## 4 METHOD

This section first introduces the GeoSSL framework and then proposes the GeoSSL-DDM algorithm. We start with exploring the coordinate perturbation for molecular data in Section 4.1. Then we introduce a coordinate-aware mutual information (MI) maximization formula and turn it into a coordinate denoising framework in Section 4.2. Nevertheless, the coordinate denoising is non-trivial since it requires geometric data reconstruction, and we adopt the score matching for estimation, as proposed in Section 4.3. The ultimate training objective is discussed in Section 4.4.

### 4.1 COORDINATE PERTURBATION FOR GEOMETRIC DATA

The mainstream self-supervised learning community designs the pretraining task by defining multiple views from the data, and these views share common information to some degree. Thus, by designing generative or contrastive tasks to maximize the mutual information (MI) between these views, the pretrained representation can encode certain key information. This will make the representation more robust and more generalizable to downstream tasks. In our work, we propose GeoSSL-DDM, an SE(3)-invariant self-supervised learning (SSL) method for molecule geometric representation learning.

The 3D geometric information or the atomic coordinates are critical to molecular properties. We carry out an additional ablation study to verify this in Appendix B. Then based on this acknowledgment, we introduce a geometry perturbation, which adds small noises to the atom coordinates. For notation, following Section 3, we define the original geometry graph and an augmented geometry graph as two views, denoted as $\boldsymbol{g}_1 = (X_1, R_1)$ and $\boldsymbol{g}_2 = (X_2, R_2)$, respectively. The augmented geometry graph can be seen as a coordinate perturbation to the original graph with the same atom types, *i.e.*, $X_2 = X_1$ and $R_2 = R_1 + \epsilon$, where $\epsilon$ is drawn from a normal distribution.

### 4.2 COORDINATE DENOISING WITH MI MAXIMIZATION FRAMEWORK: GEOSSL

The two views defined above share certain common information. By maximizing the mutual information (MI) between them, we expect that the learned representation can better capture the geometric information and is robust to noises and thus can generalize well to downstream tasks. To maximize the MI, we turn to maximize the following lower bound on the two geometry views, leading to the geometric self-supervised learning framework, GeoSSL:

$$\mathcal{L}_{\text{GeoSSL}} \triangleq \frac{1}{2}\mathbb{E}_{p(\boldsymbol{g}_1, \boldsymbol{g}_2)}\Big[\log p(\boldsymbol{g}_1|\boldsymbol{g}_2) + \log p(\boldsymbol{g}_2|\boldsymbol{g}_1)\Big]. \tag{2}$$

In Equation (2), we transform the MI maximization problem into maximizing the summation of two conditional log-likelihoods. In addition, these two conditional log-likelihoods are in the mirroring direction, and such symmetry can reveal certain nice properties, *e.g.*, it highlights the equal importance and uncertainty of the two views and can lead to a more robust representation of the geometry.

To solve Equation (2), we adopt the energy-based model (EBM) for estimation. EBM has been acknowledged as a flexible framework for its powerful usage in modeling distribution over highly-structured data, like molecules [27, 34]. To adapt it for GeoSSL, the objective can be turned into:

$$\begin{aligned}
\mathcal{L}_{\text{GeoSSL-EBM}} &= \frac{1}{2}\mathbb{E}_{p(\boldsymbol{g}_1, \boldsymbol{g}_2)}\Big[\log p(R_1|\boldsymbol{g}_2)\Big] + \frac{1}{2}\mathbb{E}_{p(\boldsymbol{g}_1, \boldsymbol{g}_2)}\Big[\log p(R_2|\boldsymbol{g}_1)\Big] \\
&= \frac{1}{2}\mathbb{E}_{p(\boldsymbol{g}_1, \boldsymbol{g}_2)}\Big[\log \frac{\exp(f(R_1, \boldsymbol{g}_2))}{A_{R_1|\boldsymbol{g}_2}}\Big] + \frac{1}{2}\mathbb{E}_{p(\boldsymbol{g}_2, \boldsymbol{g}_1)}\Big[\log \frac{\exp(f(R_2, \boldsymbol{g}_1))}{A_{R_2|\boldsymbol{g}_1}}\Big],
\end{aligned} \tag{3}$$

where the $f(\cdot)$ are the negative of energy functions, and $A_{R_1|\boldsymbol{g}_2}$ and $A_{R_2|\boldsymbol{g}_1}$ are the intractable partition functions. The first equation in Equation (3) is because the two views share the same atom types. This equation can be treated as denoising the atom coordinates of one view from the geometry of the other view. In the following, we will explore how to use the score matching for solving the above EBM estimation problem, and further transform the coordinate-aware GeoSSL to the denoising distance matching as the final objective.

### 4.3 FROM COORDINATE DENOISING TO DISTANCE DENOISING: GEOSSL-DDM

Before going into details, first, we would like to briefly discuss denoising score matching (DSM). DSM has three main advantages that inspire us to apply it for solving the coordinate-aware GeoSSL. (1) The DSM solution has a nice formulation, such that the final objective function can be simplified

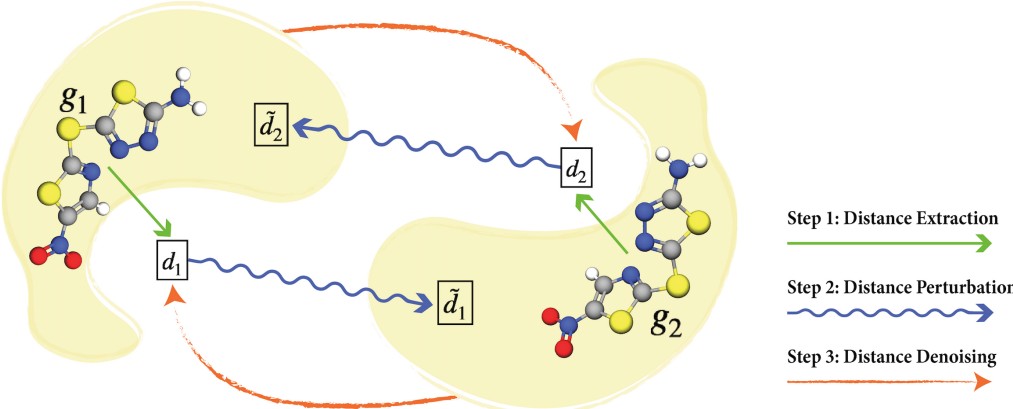

Figure 2: Pipeline for GeoSSL-DDM. The $g_1$ and $g_2$ are around the same local minima, yet with coordinate noises perturbation. Originally we want to conduct coordinate denoising between these two views. Then as proposed in GeoSSL-DDM, we transform it to an equivalent problem, *i.e.*, distance denoising. This figure shows the three key steps: extract the distances from the two geometric views, perform distance perturbation, and denoise the perturbed distances. Notice that the covalent bonds in the 3D data are added for illustration only.

with an intuitive explanation: GeoSSL-DDM can be seen as solving the denoising pairwise distance at multiple noise levels. (2) The score defined in geometric data can be viewed as a coordinate-based pseudo-force. Such pseudo-force can play an important role in the corresponding geometric representation learning. (3) In terms of the MI maximization, existing methods like InfoNCE [68], EBM-NCE, and Representation Reconstruction [38] map the data to the *representation space* for either inter-data contrastive learning or intra-data reconstruction. This operation can avoid the decoding design issue for highly-structured data [13], yet the trade-off is losing the data-inherent information by a certain degree. In other words, the data-level reconstruction task (*e.g.*, DSM) is expected to lead to a more robust representation. Thus, considering the above points, we adopt DSM to our framework and propose GeoSSL-DDM. We expect that it can learn an expressive geometric representation function by solving the coordinate-aware GeoSSL. Additionally, the two terms in Equation (3) are in the mirroring direction. Thus in what follows, we adopt a proxy task that can calculate the two directions separately, and we take one for illustration, *e.g.*, $\log \frac{\exp(f(R_1, g_2))}{A_{R_1 | g_2}}$.

### 4.3.1 DENOISING DISTANCE MATCHING

**Score.** The score is defined as the gradient of the log-likelihood w.r.t. the data, *i.e.*, the atom coordinates in our case. Because the normalization function is a constant regarding the data, it will disappear during the score calculation. To adapt it into our setting, the score is obtained as the gradient of the negative energy function w.r.t. the atom coordinates, as:

$$s(R_1, \boldsymbol{g}_2) \triangleq \nabla_{R_1} \log p(R_1 | \boldsymbol{g}_2) = \nabla_{R_1} f(R_1, \boldsymbol{g}_2). \tag{4}$$

If we assume that the learned optimal energy function, *i.e.*, $f(\cdot)$, possesses certain physical or chemical information, then the score in Equation (4) can be viewed as a special form of the pseudo-force. This may require more domain-specific knowledge, which we leave for future exploration.

**Score decomposition: from coordinates to distances.** Through back-propagation [54], the score on atom coordinates can be further decomposed into the scores attached to pairwise distances:

$$s(R_1, \boldsymbol{g}_2)_i = \sum_{j \neq i} \frac{\partial f(R_1, \boldsymbol{g}_2)}{\partial d_{1,ij}} \cdot \frac{\partial d_{1,ij}}{\partial r_{1,i}} = \sum_{j \neq i} \frac{1}{d_{1,ij}} \cdot s(\boldsymbol{d}_1, \boldsymbol{g}_2)_{ij} \cdot (r_{1,i} - r_{1,j}), \tag{5}$$

where $r_{1,i}$ is the $i$-th coordinate in $g_1$, $d_{1,ij}$ denotes the pairwise distance between the $i$-th and $j$-th nodes in $g_1$, and $s(\boldsymbol{d}_1, \boldsymbol{g}_2)_{ij} \triangleq \frac{\partial f(R_1, \boldsymbol{g}_2)}{\partial d_{1,ij}}$. Such decomposition has a nice intuition from the pseudo-force perspective: the pseudo-force on each atom can be further decomposed as the summation of pseudo-forces attached to the pairwise distances between this atom and all its neighbors. Note that here the pairwise atoms are connected in the 3D Euclidean space, not by the covalent bonds.

**Denoising distance matching (DDM).** Then we adopt the denoising score matching (DSM) [69] to our task. To be more concrete, we take the Gaussian kernel as the perturbed noise distribution on each

pairwise distance, *i.e.*, $q_\sigma(\tilde{d}_1|g_2) = \mathbb{E}_{p_{\text{data}}(d_1|g_2)}[q_\sigma(\tilde{d}_1|d_1)]$, where $\sigma$ is the deviation in Gaussian perturbation. One main advantage of using the Gaussian kernel is that the following gradient of conditional log-likelihood has a closed-form formulation: $\nabla_{\tilde{d}_1} \log q_\sigma(\tilde{d}_1|d_1, g_2) = (d_1 - \tilde{d}_1)/\sigma^2$, and the objective function of DSM is to train a score network to match it. This trick was first introduced in [69], and has been widely utilized in deep generative modeling tasks [58, 59].

To adapt to our setting, this is essentially saying that we want to train a score network, *i.e.*, $s_\theta(\tilde{d}_1|g_2)$, to match the distance perturbation, or we can say it aims at matching the pseudo-force with the pairwise distances from the pseudo-force aspect. By taking the Fisher divergence as the discrepancy metric and the trick mentioned above, the estimation objective can be simplified to

$$D_F(q_\sigma(\tilde{d}_1|g_2)||p_\theta(\tilde{d}_1|g_2)) = \frac{1}{2}\mathbb{E}_{p_{\text{data}}(d_1|g_2)}\mathbb{E}_{q_\sigma(\tilde{d}_1|d_1,g_2)}\big[\|s_\theta(\tilde{d}_1,g_2) - \frac{d_1 - \tilde{d}_1}{\sigma^2}\|^2\big] + C. \quad (6)$$

For more detailed derivations, please refer to Appendix C. In this section, we turn the coordinate-aware GeoSSL framework into a distance perturbation matching problem, which is equivalent to denoising distance matching, *i.e.*, GeoSSL-DDM. The corresponding pipeline is illustrated in Figure 2.

### 4.3.2 SE(3)-INVARIANT SCORE NETWORK MODELING

The objective function in Equation (6) is essentially doing the distance denoising. Since the distance is a type-0 feature [65], we simply design an SE(3)-invariant score network as $s_\theta(\cdot)$. For modeling $h(\cdot)$, we take an SE(3)-equivariant 3D geometric graph neural network as the geometric representation backbone model. Following the notations in Section 3 and $g_2$ modeling, we have

$$h(g_2)_i = \text{3D-GNN}(T(g_2))_i, \qquad h(g_2)_{ij} = h(g_2)_i + h(g_2)_j, \quad (7)$$

for the atom-level and atom pairwise-level representation. Then we define the score network as:

$$s_\theta(\tilde{d}_1, g_2)_{ij} = \text{MLP}\big(\text{MLP}(\tilde{d}_{1,ij}) \oplus h(g_2)_{ij}\big), \quad (8)$$

where $\oplus$ is the concatenation and MLP is the multi-layer perception. GeoSSL-DDM is agnostic to the backbone geometric representation function, and its main module is the score network in Equation (8). Thus, GeoSSL-DDM is an SE(3)-invariant [19] pretraining algorithm. Meanwhile, the type-0 distance can be modeled in a more expressive SE(3)-equivariant manner, and we leave that for future work.

### 4.4 ULTIMATE OBJECTIVE

With the above score network modeling, we can formulate the ultimate objective function. We adopt the following four training tricks from [38, 58, 59] to stabilize the score matching training process. (1) We carry out the distance denoising at $L$-level of noises. (2) We add a weighting coefficient $\lambda(\sigma) = \sigma^\beta$ for each noise level, where $\beta$ acts as the annealing factor. (3) We scale the score network by a factor of $1/\sigma$. (4) We sample the same atoms from the two geometry views with a masking ratio $r$. Ultimately, the objective function for GeoSSL-DDM, is as follows:

$$\mathcal{L}_{\text{GeoSSL-DDM}} = \frac{1}{2L}\sum_{l=1}^{L}\sigma_l^\beta \mathbb{E}_{p_{\text{data}}(d_1|g_2)}\mathbb{E}_{q(\tilde{d}_1|d_1,g_2)}\Big[\Big\|\frac{s_\theta(\tilde{d}_1,g_2)}{\sigma_l} - \frac{d_1 - \tilde{d}_1}{\sigma_l^2}\Big\|_2^2\Big]$$
$$+ \frac{1}{2L}\sum_{l=1}^{L}\sigma_l^\beta \mathbb{E}_{p_{\text{data}}(d_2|g_1)}\mathbb{E}_{q(\tilde{d}_2|d_2,g_1)}\Big[\Big\|\frac{s_\theta(\tilde{d}_2,g_1)}{\sigma_l} - \frac{d_2 - \tilde{d}_2}{\sigma_l^2}\Big\|_2^2\Big]. \quad (9)$$

The algorithm is in Algorithm 1.

**Comparison with score matching in generative modeling.** We note that score matching has been widely used for generative modeling tasks. One of the main drawbacks in the generative setting is the long mixing time for MCMC sampling. However, our work aims at representation

---

**Algorithm 1** GeoSSL-DDM pretraining

1: **Input:** A 3D geometry dataset and $L$ levels of Gaussian noise.
2: **Output:** A pre-trained 3D representation function $h(\cdot)$.
3: **for** each 3D geometry graph $g_1$ **do**
4:     Obtain $g_2$ by adding Gaussian noises to atom coordinates in $g_1$.
5:     **for** each noise level $l \in \{1, ..., L\}$ **do**
6:         Add noise to the pairwise distance with $\tilde{d}_1 = d_1 + \sigma_l, \tilde{d}_2 = d_2 + \sigma_l$.
7:         Get the score $s_\theta(\tilde{d}_1, g_2), s_\theta(\tilde{d}_2, g_1)$ with Equation (8) accordingly.
8:     **end for**
9:     Update 3D GNN representation function $h(\cdot)$ using Equation (9).
10: **end for**

---

learning, so such a sampling issue will not affect our task. We further note that there also exists a series of works exploring the score matching for conformation generation [54]. However, their scores or pseudo-forces are attached to the 2D topology (the covalent bonds), while our work is for the pure geometric data and is attached to the pairwise distances defined in the 3D Euclidean space.

## 5 EXPERIMENTS

In this section, we compare our method with nine 3D geometric pretraining baselines, including one randomly initialized, one supervised, and seven self-supervised approaches. For the downstream tasks, we adopt 22 tasks covering quantum mechanics prediction, force prediction, and binding affinity prediction. We provide all the experiment details and ablation studies in Appendix D.

### 5.1 BACKBONE MODELS

Our proposed GeoSSL-DDM is model-agnostic, and here we evaluate our method using one of the state-of-the-art geometric graph neural networks, PaiNN [51]. We carry out the exact same experiments on another backbone model, SchNet [49], and present the results in Appendix D.

PaiNN [51] is a follow-up work of SchNet [49]. It addresses the limitation of rotational equivariance in SchNet by embracing rotational invariance, attaining a more expressive 3D geometric model.

**Other backbone models.** First, we want to highlight that what we propose is a general solution and is agnostic to the backbone 3D geometric models. And in addition to the PaiNN model, we want to acknowledge that, recently, there have been several works along this research line, including but not limited to [6, 17, 17, 32, 41, 47, 56]. Yet, they may require large computation resources and may be infeasible (*e.g.*, out of GPU memory) in our setting. The decision is made by considering the model performance, computation efficiency, and memory cost. For more benchmark results and detailed comparisons of the 3D geometric models, please check Appendix A.

### 5.2 BASELINES AND PRETRAINING DATASET

**Pretraining dataset.** The PubChemQC database is a large-scale database with around 4M molecules with 3D geometries, and it calculates both the ground-state and excited-state 3D geometries using DFT (density functional theory). Due to the high computational cost, only several thousand molecules can be processed every day, and this dataset takes years of effort in total. Following this, Molecule3D [73] takes the ground-state geometries from PubChemQC and transforms the data formats into a deep learning-friendly way. It also parses essential quantum properties for each molecule, including energies of the highest occupied molecular orbital (HOMO) and the lowest occupied molecular orbital (LUMO), the energy gap between HOMO-LUMO, and the total energy. For our molecular geometry pretraining, we take a subset of 1M molecules with 3D geometries from Molecule3D.

**Self-supervised learning pretraining baselines.** We first consider the four coordinate-MI-unaware SSL methods: (1) *Type Prediction* is to predict the atom type of masked atoms; (2) *Distance Prediction* aims to predict the pairwise distances among atoms; (3) *Angle Prediction* is to predict the angle among triplet atoms, *i.e.*, the bond angle prediction; (4) *3D InfoGraph* adopts the contrastive learning paradigm by taking the node-graph pair from the same molecule geometry as positive and negative otherwise. Next, following the coordinate-aware GeoSSL framework introduced in Equation (2), we include two contrastive and one generative SSL baselines. (5) *GeoSSL-InfoNCE* [68] and (6) *GeoSSL-EBM-NCE* [38] are the two widely-used contrastive learning loss functions, where the goal is to align the positive views and contrast the negative views simultaneously. Finally, (7) *GeoSSL-RR* (RR for Representation Reconstruction) [38] is a generative SSL that is a proxy to maximize the MI. RR is a more general form of non-contrastive SSL methods like BOYL [23] and SimSiam [9], and the goal is to reconstruct each view from its counterpart in the representation space. Following this, our proposed GeoSSL-DDM, can be classified as generative SSL for distance denoising.

**Supervised pretraining baseline.** We also compare our method with a supervised pretraining baseline. As aforementioned, the large-scale pretraining dataset uses the DFT to calculate the energy and extracts the most stable conformers with the lowest energies, which reveal the most fundamental properties of molecules in the 3D Euclidean space. Thus, such energies can be naturally adopted as supervised signals, and we take this as a supervised pretraining baseline.

### 5.3 DOWNSTREAM TASKS ON QUANTUM MECHANICS AND FORCE PREDICTION

QM9 [46] is a dataset of 134K molecules consisting of 9 heavy atoms. It includes 12 tasks that are related to the quantum properties. For example, U0 and U298 are the internal energies at 0K at 0K and 298.15K respectively, and U298 and G298 are the other two energies that can be transferred from

Table 1: Downstream results on 12 quantum mechanics prediction tasks from QM9. We take 110K for training, 10K for validation, and 11K for test. The evaluation is mean absolute error, and the best results are in **bold**.

| Pretraining | Alpha ↓ | Gap ↓ | HOMO↓ | LUMO ↓ | Mu ↓ | Cv ↓ | G298 ↓ | H298 ↓ | R2 ↓ | U298 ↓ | U0 ↓ | Zpve ↓ |
|---|---|---|---|---|---|---|---|---|---|---|---|---|
| – | 0.048 | 44.50 | 26.00 | 21.11 | 0.016 | 0.025 | 8.31 | 7.67 | 0.132 | 7.77 | 7.89 | 1.322 |
| Supervised | 0.049 | 45.33 | 26.61 | 21.77 | 0.016 | 0.026 | 8.97 | 8.59 | 0.170 | 8.35 | 8.19 | 1.346 |
| Type Prediction | 0.050 | 47.28 | 30.56 | 23.18 | 0.016 | **0.024** | 9.32 | 9.10 | 0.163 | 8.94 | 8.60 | 1.357 |
| Distance Prediction | 0.063 | 47.62 | 29.18 | 22.40 | 0.019 | 0.045 | 12.02 | 12.31 | 0.636 | 11.76 | 12.22 | 1.840 |
| Angle Prediction | 0.056 | 47.36 | 29.53 | 22.61 | 0.018 | 0.027 | 10.23 | 10.13 | 0.143 | 9.95 | 9.70 | 1.643 |
| 3D InfoGraph | 0.053 | 44.79 | 27.09 | 21.66 | 0.016 | 0.027 | 9.22 | 8.78 | 0.143 | 8.94 | 9.11 | 1.465 |
| GeoSSL-RR | 0.048 | 44.85 | 25.42 | 20.82 | **0.015** | 0.025 | 8.56 | 8.20 | 0.133 | 7.89 | 7.62 | 1.329 |
| GeoSSL-InfoNCE | 0.052 | 45.65 | 26.70 | 21.87 | 0.016 | 0.027 | 9.17 | 9.62 | 0.130 | 8.77 | 8.63 | 1.519 |
| GeoSSL-EBM-NCE | 0.049 | 44.18 | 26.29 | 21.46 | **0.015** | 0.026 | 8.56 | 8.13 | 0.126 | 8.01 | 7.96 | 1.447 |
| GeoSSL-DDM (ours) | **0.046** | **40.22** | **23.48** | **19.42** | **0.015** | **0.024** | **7.65** | **7.09** | **0.122** | **6.99** | **6.92** | **1.307** |

Table 2: Downstream results on 8 force prediction tasks from MD17. We take 1K for training, 1K for validation, and the number of molecules for test are varied among different tasks, ranging from 48K to 991K. The evaluation is mean absolute error, and the best results are in **bold**.

| Pretraining | Aspirin ↓ | Benzene ↓ | Ethanol ↓ | Malonaldehyde ↓ | Naphthalene ↓ | Salicylic ↓ | Toluene ↓ | Uracil ↓ |
|---|---|---|---|---|---|---|---|---|
| – | 0.556 | 0.052 | 0.213 | 0.338 | 0.138 | 0.288 | 0.155 | 0.194 |
| Supervised | 0.478 | 0.145 | 0.318 | 0.434 | 0.460 | 0.527 | 0.251 | 0.404 |
| Type Prediction | 1.656 | 0.349 | 0.414 | 0.886 | 1.684 | 1.807 | 0.660 | 1.020 |
| Distance Prediction | 1.434 | 0.090 | 0.378 | 1.017 | 0.631 | 1.569 | 0.350 | 0.415 |
| Angle Prediction | 0.839 | 0.105 | 0.337 | 0.517 | 0.772 | 0.931 | 0.274 | 0.676 |
| 3D InfoGraph | 0.844 | 0.114 | 0.344 | 0.741 | 1.062 | 0.945 | 0.373 | 0.812 |
| GeoSSL-RR | 0.502 | 0.052 | 0.219 | 0.334 | 0.130 | 0.312 | 0.152 | 0.192 |
| GeoSSL-InfoNCE | 0.881 | 0.066 | 0.275 | 0.550 | 0.356 | 0.607 | 0.186 | 0.559 |
| GeoSSL-EBM-NCE | 0.598 | 0.073 | 0.237 | 0.518 | 0.246 | 0.416 | 0.178 | 0.475 |
| GeoSSL-DDM (ours) | **0.453** | **0.051** | **0.166** | **0.288** | **0.129** | **0.266** | **0.122** | **0.183** |

H298 respectively. The other 8 tasks are quantum mechanics related to the DFT process. MD17 [10] is a dataset on molecular dynamics simulation. It includes eight tasks, corresponding to eight organic molecules, and each task includes the molecule positions along the potential energy surface (PES), as shown in Figure 1. The goal is to predict the energy-conserving interatomic forces for each atom in each molecule position. We follow the literature [31, 41, 50, 51] of using 1K for training and 1K for validation, while the test set (from 48K to 991K) is much larger.

The results on QM9 and MD17 are displayed in Tables 1 and 2 respectively. From Tables 1 and 2, we can observe that most the pretraining baselines tested perform on par with or even worse than the randomly-initialized baseline. The top performing baseline is the representation reconstruction method (RR), which optimizes the coordinate-aware MI; it outperforms the other baselines on 5 out of 12 tasks in QM9 and 6 out of 8 tasks in MD17. This implies the potential of applying generative SSL for maximizing this coordinate-aware MI. Promisingly, our proposed GeoSSL-DDM, achieves consistently improved performance on all 12 tasks in QM9 and 8 tasks in MD17. All these observations empirically verify the effectiveness of the distance denoising in GeoSSL-DDM, which models the most determinant factor in molecule geometric data.

## 5.4 Downstream Tasks on Binding Affinity Prediction

Atom3D [66] is a recently published dataset. It gathers several core tasks for 3D molecules, including binding affinity. The binding affinity prediction is to measure the strength of binding interaction between a small molecule to the target protein. Here we will model both the small molecule and protein with their 3D atom coordinates provided. We follow Atom3D in data preprocessing and data splitting. For more detailed discussions and statistics, please check Appendix D.

During the binding process, there is a cavity in a protein that can potentially possess suitable properties for binding a small molecule (ligand), and it is termed a pocking [62]. Because of the large volume of the protein, we follow [66] by only taking the binding pocket, where there are no more than 600 atoms for each molecule and protein pair. To be more concrete, we consider two binding affinity tasks. (1) The first task is ligand binding affinity (LBA). It is gathered from [70] and the task is to predict the binding affinity strength between a small molecule and a protein pocket. (2) The second task is ligand efficacy prediction (LEP). We have a molecule bounded to pockets, and the goal is to detect if the same molecule has a higher binding affinity with one pocket compared to the other one.

Results in Table 3 illustrate that, for the LBA task, two pretraining baseline methods fail to generalize to LBA (the loss gets too large), and all the other pretraining baselines cannot beat the randomly initialized baseline. For the LEP task, the supervised and two contrastive learning pretraining

Table 3: Downstream results on 2 binding affinity tasks. We select three evaluation metrics for LBA: the root mean squared error (RMSD), the Pearson correlation ($R_p$) and the Spearman correlation ($R_S$). LEP is a binary classification task, and we use the area under the curve for receiver operating characteristics (ROC) and precision-recall (PR) for evaluation. We run cross validation with 5 seeds, and the best results are in **bold**.

| Pretraining | LBA | | | LEP | |
|---|---|---|---|---|---|
| | RMSD ↓ | $R_P$ ↑ | $R_C$ ↑ | ROC ↑ | PR ↑ |
| – | $1.463 \pm 0.06$ | $0.572 \pm 0.02$ | $0.568 \pm 0.02$ | $0.675 \pm 0.04$ | $0.549 \pm 0.05$ |
| Supervised | $1.551 \pm 0.08$ | $0.539 \pm 0.03$ | $0.533 \pm 0.03$ | $0.696 \pm 0.03$ | $0.554 \pm 0.03$ |
| Charge Prediction | $2.316 \pm 0.80$ | $0.387 \pm 0.11$ | $0.400 \pm 0.11$ | $0.630 \pm 0.05$ | $0.557 \pm 0.07$ |
| Distance Prediction | $1.542 \pm 0.08$ | $0.545 \pm 0.03$ | $0.540 \pm 0.03$ | $0.521 \pm 0.07$ | $0.479 \pm 0.07$ |
| Angle Prediction | – | – | – | $0.545 \pm 0.07$ | $0.504 \pm 0.07$ |
| 3D InfoGraph | – | – | – | $0.540 \pm 0.03$ | $0.469 \pm 0.03$ |
| GeoSSL-RR | $1.515 \pm 0.07$ | $0.545 \pm 0.03$ | $0.539 \pm 0.03$ | $0.654 \pm 0.05$ | $0.518 \pm 0.06$ |
| GeoSSL-InfoNCE | $1.564 \pm 0.05$ | $0.508 \pm 0.03$ | $0.497 \pm 0.05$ | $0.693 \pm 0.06$ | $0.571 \pm 0.08$ |
| GeoSSL-EBM-NCE | $1.499 \pm 0.06$ | $0.547 \pm 0.03$ | $0.534 \pm 0.03$ | $0.691 \pm 0.05$ | $0.603 \pm 0.07$ |
| GeoSSL-DDM (ours) | $\mathbf{1.451 \pm 0.03}$ | $\mathbf{0.577 \pm 0.02}$ | $\mathbf{0.572 \pm 0.01}$ | $\mathbf{0.776 \pm 0.03}$ | $\mathbf{0.694 \pm 0.06}$ |

baselines stand out for both ROC and PR metrics. Meaningfully, for both tasks, GeoSSL-DDM is able to achieve promising improvement, revealing that modeling the local region around conformer with distance denoising can also benefit binding affinity downstream tasks.

## 5.5 DISCUSSION: CONNECTION WITH MULTI-TASK PRETRAINING

In the above experiments, we test multiple self-supervised and supervised pretraining tasks separately. Yet, all these pretraining methods are not contradicted but could be complementary instead. Existing work has successfully shown the effect of combining them in various ways. For example, [28] shows that jointly doing supervised and self-supervised pretraining can augment the pretrained representation. [38, 57] prove that contrastive and generative SSL pretraining methods can be learned simultaneously as a multi-task pretraining. In addition, in terms of the molecule-specific pretraining, [38] empirically verifies that 2D topology and 3D geometry views can share certain information, and maximizing their mutual information together with 2D topology SSL for pretraining is beneficial.

With these insights, we would like to claim that all of these points are worth exploring in the future, especially in the line of pretraining for molecular geometry. Because pretraining datasets often come with multiple quantum properties and the 2D molecular topology can be obtained heuristically. Yet as the first step to explore self-supervised learning using only the 3D geometric data (*i.e.*, without covalent bonds), our study here would like to leave multi-task pretraining for future exploration.

## 6 CONCLUSIONS AND FUTURE DIRECTIONS

We proposed a novel coordinate denoising method, coined GeoSSL-DDM, for molecular geometry pretraining. GeoSSL-DDM leverages an SE(3)-invariant score matching strategy, under the GeoSSL framework, to successfully decompose its coordinate denoising objective into the denoising of pairwise atomic distances in a molecule, which then can be effectively computed and directly target the determinant factors in molecular geometric data. We empirically verified the effectiveness and robustness of our method, showing its superior performance to nine state-of-the-art pretraining baselines on 22 benchmarking geometric molecular property prediction and binding affinity tasks.

Our work opens up venues for multiple promising directions. First, from the machine learning perspective, we propose a general pipeline on using EBM for MI maximization on geometric data pretraining. Yet, there are more explorations on the success of EBM, like GFlowNet [3], and it would be interesting to explore how to combine it with molecular geometric data along this systematic path. In addition, GeoSSL-DDM does not utilize the 2D structure (*i.e.*, covalent bonds for molecules), and it would be desirable to consider how to utilize the distance denoising together with the 2D topology information.

In terms of applications, our proposed GeoSSL-DDM is a general framework, and it can be naturally applied to other geometric data, such as point clouds and protein pretraining. In addition, our current goal is to perform denoising in the local region, yet it would be interesting to explore larger regions. From this aspect, the denoising can be viewed as recovering the molecular dynamics trajectory, and we would explore how generalizable this pretrained representation is to downstream tasks.

## ETHICS STATEMENT

We authors acknowledge that we have read and committed to adhering to the ICLR Code of Ethics.

## REPRODUCIBILITY STATEMENT

To ensure the reproducibility of the empirical results, we provide the implementation details (hyper-parameters, dataset statistics, etc.) in Section 5 and appendix D, and publically share our source code through this GitHub link. Besides, the complete derivations of equations and clear explanations are shown in Section 4 and appendix C.

## ACKNOWLEDGEMENT

We would like to thank Anima Anandkumar, Chaowei Xiao, Weili Nie, Zhuoran Qiao, Chengpeng Wang, and Pierre-André Noël for their insightful discussions. This project is supported by the Natural Sciences and Engineering Research Council (NSERC) Discovery Grant, the Canada CIFAR AI Chair Program, collaboration grants between Microsoft Research and Mila, Samsung Electronics Co., Ltd., Amazon Faculty Research Award, Tencent AI Lab Rhino-Bird Gift Fund and two NRC Collaborative R&D Projects (AI4D-CORE-06, AI4D-CORE-08). This project was also partially funded by IVADO Fundamental Research Project grant PRF-2019-3583139727.

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

# A    BENCHMARKS AND RELATED WORK

## A.1    GEOMETRIC NEURAL NETWORKS

Recently, geometric neural networks have been actively proposed, including SchNet [50], TFN [17], DimeNet++ [31], SE(3)-Trans [17], EGNN [47], SEGNN [6], SphereNet [41], SpinConv [56], PaiNN [51], and GemNet [32]. We reproduce most of them on the QM9 dataset as shown in Table 4. Among this, we would like to highlight two models: SchNet and PaiNN.

**SchNet** [49] is composed of the following key steps:

$$z_i^{(0)} = \text{embedding}(x_i), \quad z_i^{(t+1)} = \text{MLP}\Big(\sum_{j=1}^n f(x_j^{(t-1)}, r_i, r_j)\Big), \quad h_i = \text{MLP}(z_i^{(K)}), \qquad (10)$$

where $K$ is the number of hidden layers, and

$$f(x_j, r_i, r_j) = x_j \cdot e_k(r_i - r_j) = x_j \cdot \exp(-\gamma \|\|r_i - r_j\|_2 - \mu\|_2^2) \qquad (11)$$

is the continuous-filter convolution layer, enabling the modeling of continuous coordinates of atoms.

**PaiNN** [51] is an improved work of SchNet [49]. It addresses the limitation of rotational equivariance in SchNet by embracing rotational invariance, attaining a more expressive SE(3)-equivariant neural network model.

## A.2    BENCHMARK ON QM9

Current work is using different optimization strategies and different data split (in terms of the splitting size). Originally there are 133,885 molecules in QM9, where 3,054 are filtered out, leading to 130,831 molecules. During the benchmark, we find that:

- The performance on QM9 is very robust to either using (1) 110K for training, 10K for val, 10,831 for test or using (2) 100K for training, 13,083 for val and 17,748 for test.
- The optimization, especially the learning rate scheduler is very critical. During the benchmarking, we find that using cosine annealing learning rate schedule [43] is generally the most robust.

For more detailed discussion on QM9, please refer to Appendix D. We show the benchmark results on QM9 in Table 4.

Table 4: Benchmark results on 12 quantum mechanics prediction tasks from QM9. We take 110K for training, 10K for validation, and 11K for test. The evaluation is mean absolute error (MAE).

|  | Alpha ↓ | Gap ↓ | HOMO↓ | LUMO ↓ | Mu ↓ | Cv ↓ | G298 ↓ | H298 ↓ | R2 ↓ | U298 ↓ | U0 ↓ | Zpve ↓ |
|---|---|---|---|---|---|---|---|---|---|---|---|---|
| SchNet | 0.070 | 50.38 | 31.81 | 25.76 | 0.029 | 0.031 | 14.60 | 14.24 | 0.131 | 13.99 | 14.12 | 1.686 |
| SE(3)-Trans | 0.136 | 58.27 | 35.95 | 35.41 | 0.052 | 0.068 | 68.50 | 70.22 | 1.828 | 70.14 | 72.28 | 5.302 |
| EGNN | 0.067 | 48.77 | 28.98 | 24.44 | 0.032 | 0.031 | 11.02 | 11.07 | 0.078 | 10.83 | 10.70 | 1.578 |
| DimeNet++ | 0.046 | 38.14 | 21.23 | 17.57 | 0.029 | 0.022 | 7.98 | 7.19 | 0.306 | 6.86 | 6.93 | 1.204 |
| SphereNet | 0.050 | 39.54 | 21.88 | 18.66 | 0.026 | 0.025 | 8.65 | 7.43 | 0.262 | 8.28 | 8.01 | 1.390 |
| SEGNN | 0.057 | 41.08 | 22.46 | 21.46 | 0.025 | 0.028 | 13.07 | 13.94 | 0.472 | 14.64 | 13.89 | 1.662 |
| PaiNN | 0.048 | 44.50 | 26.00 | 21.11 | 0.016 | 0.025 | 8.31 | 7.67 | 0.132 | 7.77 | 7.89 | 1.322 |

## A.3    RELATED WORK

We acknowledge that there is a parallel work called Protein Tertiary SSL (PTSSL) [24] working on the geometric self-supervised learning. Yet, there are some fundamental differences between theirs and ours, as listed below: **(1) Key notion on pseudo-force.** PTSSL directly applies the denoised score matching method into protein tertiary structures, yet our focus is on how the notion of pseudo-force can come into the play, which possess better generalization ability. **(2) Task setting.** PTSSL works on protein and utilize both the 2D and 3D information, and our work is purely working on the 3D geometric information. **(3) Technical novelty.** PTSSL designs the DSM objective for SSL, and what we propose is a systematic tool: using energy-based model and score matching to solve the geometric SSL problem opens a new venue in this field. **(4) Objective.** PTSSL directly designs one objective function, which is denoising from one view to the other. Ours starts from the

lower bound of MI, which is symmetric in terms of the denoising directions. We believe that such symmetry are treating the two views equally, and can better reveal the mutual concept, making the pre-trained representation more robust to the position augmentations. **(5) Empirical baseline.** PTSSL lacks the comparisons with other pre-training methods, while we compare with 7 SOTA pre-training methods, especially those driven by maximizing the MI with the same augmentations. Without such comparisons, it is hard to tell the effectiveness of the pseudo-force matching for geometric data. **(6) Score network.** Last but not least, the score network designed in PTSSL does not satisfy the SE(3) equivariant property.

## B   AN EXAMPLE ON THE IMPORTANCE OF ATOM COORDINATES

First, it has been widely acknowledged [15] that the atom positions or molecule shapes are important factors to the quantum properties. Here we carry out an evidence example to empirically verify this. The goal here is to make predictions on 12 quantum properties in QM9.

The molecule geometric data includes two main components as input features: the atom types and atom coordinates. Other key information can be inferred accordingly, including the pairwise distances and torsion angles. We consider corruption in each of the components to empirically test their importance accordingly.

- Atom type corruption. There are in total 118 types of atom types, and the standard embedding option is to apply the one-hot encoding. In the corruption case, we replace all the atom types with a hold-out index, *i.e.*, index 119.
- Atom coordinate corruption. Originally QM9 includes atom coordinates that are in the stable state, and now we replace them with the coordinates generated with MMFF [26] from RDKit [33].

Table 5: An evidence example on molecular data. The goal is to predict 12 quantum properties (regression tasks) of 3D molecules (with 3D coordinates on each atom). The evaluation metric is MAE.

| Model | Mode | Alpha ↓ | Gap ↓ | HOMO↓ | LUMO ↓ | Mu ↓ | Cv ↓ | G298 ↓ | H298 ↓ | R2 ↓ | U298 ↓ | U0 ↓ | Zpve ↓ |
|---|---|---|---|---|---|---|---|---|---|---|---|---|---|
| SchNet | Stable Geometry | 0.070 | 50.59 | 32.53 | 26.33 | 0.029 | 0.032 | 14.68 | 14.85 | 0.122 | 14.70 | 14.44 | 1.698 |
|  | Type Corruption | 0.074 | 52.07 | 33.64 | 26.75 | 0.032 | 0.032 | 21.68 | 22.93 | 0.231 | 23.01 | 22.99 | 1.677 |
|  | Coordinate Corruption | 0.265 | 110.59 | 79.92 | 78.59 | 0.422 | 0.113 | 57.07 | 58.92 | 18.649 | 60.71 | 59.32 | 5.151 |
| PaiNN | Stable Geometry | 0.048 | 44.50 | 26.00 | 21.11 | 0.016 | 0.025 | 8.31 | 7.67 | 0.132 | 7.77 | 7.89 | 1.322 |
|  | Type Corruption | 0.057 | 45.61 | 27.22 | 22.16 | 0.016 | 0.025 | 11.48 | 11.60 | 0.181 | 11.15 | 10.89 | 1.339 |
|  | Coordinate Corruption | 0.223 | 108.31 | 73.43 | 72.35 | 0.391 | 0.095 | 48.40 | 51.82 | 16.828 | 51.43 | 48.95 | 4.395 |

We take SchNet and PaiNN as the backbone 3D GNN models, and the results are in Table 5. We can observe that (1) Both corruption examples lead to performance decrease. (2) The atom coordinate corruption may lead to more severe performance decrease than the atom type corruption. To put this into another way is that, when we corrupt the atom types with the same hold-out type, it is equivalently to removing the atom type information. Thus, this can be viewed as using the equilibrium atom coordinates alone, and the property prediction is comparatively robust. This observation can also be supported from the domain perspective. According to the valence bond theory, the atom type information can be implicitly and roughly inferred from the atom coordinates.

Therefore, by combining all the above observations and analysis, one can draw the conclusion that, *for molecule geometry data, the atom coordinates reveal more fundamental information for representation learning*.

## C   MUTUAL INFORMATION MAXIMIZATION WITH ENERGY-BASED MODEL

In this section, we will give a detailed discussion on mutual information (MI) maximization with the energy-based model (EBM).

First, we can get a lower bound of MI. Assuming that there exist (possibly negative) constants $a$ and $b$ such that $a \leq H(X)$ and $b \leq H(Y)$, *i.e.*, the lower bounds to the (differential) entropies, then we have:

$$
\begin{aligned}
I(X;Y) &= \frac{1}{2}\big(H(X) + H(Y) - H(Y|X) - H(X|Y)\big) \\
&\geq \frac{1}{2}\big(a + b - H(Y|X) - H(X|Y)\big) \\
&\geq \frac{1}{2}(a + b) + \mathcal{L}_{\text{MI}},
\end{aligned}
\tag{12}
$$

where the loss $\mathcal{L}_{\text{MI}}$ is defined as:

$$
\mathcal{L}_{\text{MI}} = \frac{1}{2}\mathbb{E}_{p(\boldsymbol{x},\boldsymbol{y})}\Big[ \log p(\boldsymbol{x}|\boldsymbol{y}) + \log p(\boldsymbol{y}|\boldsymbol{x})\Big].
\tag{13}
$$

Empirically, we use energy-based models to model the distributions. The existence of $a$ and $b$ can be understood as the requirements that the two distributions $(p_{\boldsymbol{x}}, p_{\boldsymbol{y}})$ are not collapsed. Notice that to keep consistent with the notations in Section 3, we will be using $\boldsymbol{g}_1$ and $\boldsymbol{g}_2$ as the two variables. Then the goal is equivalent to optimizing the following equation:

$$
\mathcal{L}_{\text{GeoSSL}} \triangleq \frac{1}{2}\mathbb{E}_{p(\boldsymbol{g}_1,\boldsymbol{g}_2)}\Big[ \log p(\boldsymbol{g}_1|\boldsymbol{g}_2) + \log p(\boldsymbol{g}_2|\boldsymbol{g}_1)\Big].
\tag{14}
$$

Thus, we transform the MI maximization problem into maximizing the summation of two conditional log-likelihoods. Such an objective function opens a wider venue for estimating MI, *e.g.*, using the EBM to estimate Equation (14).

**Adaptation to Geometric Data**   The 3D geometric information or the atomic coordinates are critical to molecular properties. Then based on this, we propose a geometry perturbation, which adds small noises to the atom coordinates. This geometry perturbation possesses certain motivations from both domain and machine learning perspectives. (1) From the practical experiment perspective, the statistical and systematic errors [11] on conformation estimation are unavoidable. Coordinate perturbation is a natural way to enable learning representations robust to such noises. (2) From the domain aspect, molecules are not static but in continuous motion in the 3D Euclidean space, and we can obtain a potential energy surface accordingly. We are interested in modeling the conformer, *i.e.*, the 3D coordinates with the lowest energy. However, even the conformer at the lowest energy point can have vibrations, and coordinate perturbation can better capture such movement yet with the same order of magnitude on energies. (3) As will be illustrated later, our proposed method can be simplified as denoising atomic distance matching. (4) Leveraging coordinate perturbation for model regularization has also been empirically verified its effectiveness for supervised molecule geometric representation learning [21]. Such characteristics of molecular geometry motivate us to apply coordinate perturbation. If we take each of the two views as adding noise to the coordinates from the other view, then the objective in Equation (14) essentially states that we want to conduct coordinate denoising, as shown in Figure 3. Yet, this is not a trivial task due to the complicated geometric space (*e.g.*, 3D coordinates) reconstruction.

### C.1   AN EBM FRAMEWORK FOR MI ESTIMATION

The lower bound in Equation (14) is composed of two conditional log-likelihood terms, and then we model the conditional likelihood with EBM. This gives us:

$$
\mathcal{L}_{\text{GeoSSL-EBM}} = -\frac{1}{2}\mathbb{E}_{p(\boldsymbol{g}_1,\boldsymbol{g}_2)}\Big[ \log \frac{\exp(f_{\boldsymbol{g}_1}(\boldsymbol{g}_1,\boldsymbol{g}_2))}{A_{\boldsymbol{g}_1|\boldsymbol{g}_2}} + \log \frac{\exp(f_{\boldsymbol{g}_2}(\boldsymbol{g}_2,\boldsymbol{g}_1))}{A_{\boldsymbol{g}_2|\boldsymbol{g}_1}}\Big],
\tag{15}
$$

where $f_{\boldsymbol{g}_1}(\boldsymbol{g}_1,\boldsymbol{g}_2) = -E(\boldsymbol{g}_1|\boldsymbol{g}_2)$ and $f_{\boldsymbol{g}_2}(\boldsymbol{g}_2,\boldsymbol{g}_1) = -E(\boldsymbol{g}_2|\boldsymbol{g}_1)$ are the negative energy functions, and $A_{\boldsymbol{g}_1|\boldsymbol{g}_2}$ and $A_{\boldsymbol{g}_2|\boldsymbol{g}_1}$ are the corresponding partition functions. The energy functions can be flexibly defined, thus the bottleneck here is the intractable partition function due to the high cardinality. To solve this, existing methods include noise-contrastive estimation (NCE) [25] and score matching (SM) [60, 61], and we will describe how to apply them for MI maximization.

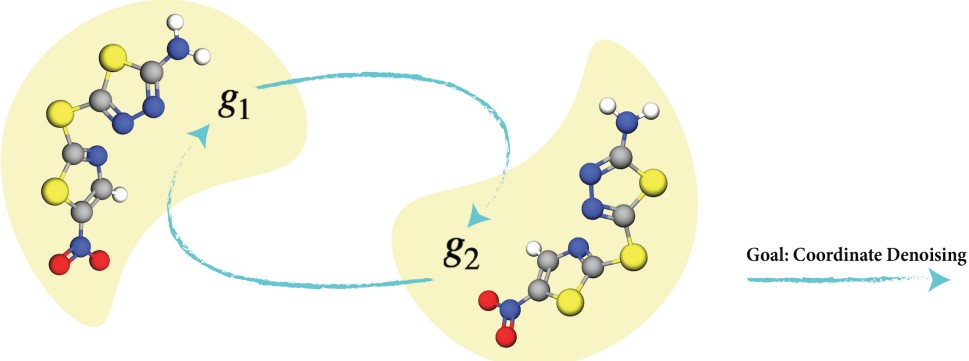

Figure 3: Pipeline for denoising coordinate matching.

## C.2 EBM-NCE FOR MI ESTIMATION

Under the EBM framework, if we solve Equation (15) with Noise-Contrastive Estimation (NCE) [25], the final objective is termed EBM-NCE, as:

$$
\mathcal{L}_{\text{GeoSSL-EBM-NCE}} = -\frac{1}{2} \mathbb{E}_{p_{\text{data}}(y)} \Big[ \mathbb{E}_{p_n(\boldsymbol{g_1}|\boldsymbol{g_2})}[\log\big(1 - \sigma(f_{\boldsymbol{g_1}}(\boldsymbol{g_1}, \boldsymbol{g_2}))\big)] + \mathbb{E}_{p_{\text{data}}(\boldsymbol{g_1}|\boldsymbol{g_2})}[\log \sigma(f_{\boldsymbol{g_1}}(\boldsymbol{g_1}, \boldsymbol{g_2}))] \Big]
$$
$$
- \frac{1}{2} \mathbb{E}_{p_{\text{data}}(x)} \Big[ \mathbb{E}_{p_n(\boldsymbol{g_2}|\boldsymbol{g_1})}[\log\big(1 - \sigma(f_{\boldsymbol{g_2}}(\boldsymbol{g_2}, \boldsymbol{g_1}))\big)] + \mathbb{E}_{p_{\text{data}}(\boldsymbol{g_2}|\boldsymbol{g_1})}[\log \sigma(f_{\boldsymbol{g_2}}(\boldsymbol{g_2}, \boldsymbol{g_1}))] \Big].
$$
(16)

All the detailed derivations can be found in [25]. Specifically, EBM-NCE is equivalent to the Jensen-Shannon estimation for MI, while the mathematical intuitions and derivation processes are different. Besides, it also belongs to the contrastive SSL venue. That is, it aims at aligning the positive pairs and contrasting the negative pairs.

## C.3 EBM-SM FOR MI ESTIMATION: GEOSSL-DDM

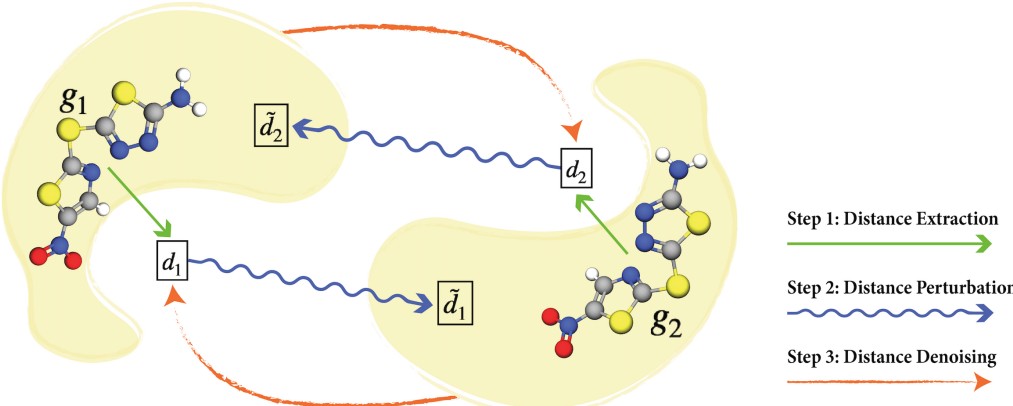

Figure 4: Pipeline for GeoSSL-DDM. The $\boldsymbol{g_1}$ and $\boldsymbol{g_2}$ are around the same local minima, yet with coordinate noises perturbation. Originally we want to do coordinate denoising between these two views. Then as proposed in GeoSSL-DDM, we transform it to an equivalent problem, *i.e.*, distance denoising. This figure shows the three key steps: extract the distances from the two geometric views, perform distance perturbation, and denoise the perturbed distances.

In this subsection, we will focus on geometric data like molecular geometry. Recall that we have two views: $\boldsymbol{g_1}$ and $\boldsymbol{g_2}$, and the goal is to maximize the lower bound of the mutual information

in Equation (14). Because the two views share the same atomic features, it can be reduced to:

$$
\begin{aligned}
\mathcal{L}_{\text{GeoSSL-EBM}} &= \frac{1}{2}\mathbb{E}_{p(\boldsymbol{g}_1,\boldsymbol{g}_2)}\Big[\log p(\boldsymbol{g}_1|\boldsymbol{g}_2)\Big] + \frac{1}{2}\mathbb{E}_{p(\boldsymbol{g}_1,\boldsymbol{g}_2)}\Big[\log p(\boldsymbol{g}_2|\boldsymbol{g}_1)\Big] \\
&= \frac{1}{2}\mathbb{E}_{p(\boldsymbol{g}_1,\boldsymbol{g}_2)}\Big[\log p(\langle X_1, R_1\rangle|\langle X_2, R_2\rangle)\Big] + \frac{1}{2}\mathbb{E}_{p(\boldsymbol{g}_1,\boldsymbol{g}_2)}\Big[\log p(\langle X_2, R_2\rangle|\langle X_1, R_1\rangle)\Big] \\
&= \frac{1}{2}\mathbb{E}_{p(\boldsymbol{g}_1,\boldsymbol{g}_2)}\Big[\log p(R_1|\boldsymbol{g}_2)\Big] + \frac{1}{2}\mathbb{E}_{p(\boldsymbol{g}_1,\boldsymbol{g}_2)}\Big[\log p(R_2|\boldsymbol{g}_1)\Big] \\
&= \frac{1}{2}\mathbb{E}_{p(\boldsymbol{g}_1,\boldsymbol{g}_2)}\Big[\log \frac{\exp(f(R_1,\boldsymbol{g}_2))}{A_{R_1|\boldsymbol{g}_2}}\Big] + \frac{1}{2}\mathbb{E}_{p(\boldsymbol{g}_2,\boldsymbol{g}_1)}\Big[\log \frac{\exp(f(R_2,\boldsymbol{g}_1))}{A_{R_2|\boldsymbol{g}_1}}\Big],
\end{aligned}
\tag{17}
$$

where the $f(\cdot)$ are the negative of energy functions, and $A_{R_1|\boldsymbol{g}_2}$ and $A_{R_2|\boldsymbol{g}_1}$ are the intractable partition functions. The first equation in Equation (17) results from that the two views share the same atom types. This equation can be treated as denoising the atom coordinates of one view from the geometry of the other view. In the following, we will explore how to use the score matching for solving EBM, and further transform the coordinate-aware mutual information maximization to the denoising distance matching (GeoSSL-DDM) as the final objective.

**Score Definition** The two terms in Equation (3) are in the mirroring direction. Thus in what follows, we may as well adopt a proxy task that these two directions can calculated separately, and take one direction for illustration, *e.g.*, $\log \frac{\exp(f(R_1,\boldsymbol{g}_2))}{A_{R_1|\boldsymbol{g}_2}}$. The score is defined as the gradient of the log-likelihood w.r.t. the data, *i.e.*, the atom coordinates in our case. Because the normalization function is a constant w.r.t. the data, it will disappear during the score calculation. To adapt it into our setting, the score is obtained as the gradient of the negative energy function w.r.t. the atom coordinates, as:

$$
s(R_1,\boldsymbol{g}_2) = \nabla_{R_1}\log p(R_1|\boldsymbol{g}_2) = \nabla_{R_1}f(R_1,\boldsymbol{g}_2).
\tag{18}
$$

If we assume that the learned optimal energy function, *i.e.*, $f(\cdot)$, possesses certain physical or chemical information, then the score in Equation (18) can be viewed as a special form of the pseudo-force. This may require more domain-specific knowledge, and we leave this for future exploration.

**Score Decomposition: From Coordinates To Distances** Through back-propagation [54], the score on atom coordinates can be further decomposed into the scores attached to pairwise distances:

$$
\begin{aligned}
s(R_1,\boldsymbol{g}_2)_i &= \frac{\partial f(R_1,\boldsymbol{g}_2)}{\partial r_{1,i}} \\
&= \sum_{j\in\mathcal{N}(i)} \frac{\partial f(R_1,\boldsymbol{g}_2)}{\partial d_{1,ij}} \cdot \frac{\partial d_{1,ij}}{\partial r_{1,i}} \\
&= \sum_{j\in\mathcal{N}(i)} \frac{1}{d_{1,ij}} \cdot \frac{\partial f(R_1,\boldsymbol{g}_2)}{\partial d_{1,ij}} \cdot (r_{1,i} - r_{1,j}) \\
&= \sum_{j\in\mathcal{N}(i)} \frac{1}{d_{1,ij}} \cdot s(\boldsymbol{d}_1,\boldsymbol{g}_2)_{ij} \cdot (r_{1,i} - r_{1,j}),
\end{aligned}
\tag{19}
$$

where $s(\boldsymbol{d}_1,\boldsymbol{g}_2)_{ij} \triangleq \frac{\partial f(R_1,\boldsymbol{g}_2)}{\partial d_{1,ij}}$. Such decomposition has a nice underlying intuition from the pseudo-force perspective: the pseudo-force on each atom can be further decomposed as the summation of pseudo-forces on the pairwise distances starting from this atom. Note that here the pairwise atoms are connected in the 3D Euclidean space, not by the covalent-bonding.

**Denoising Distance Matching (DDM)** Then we adopt the denoising score matching (DSM) [69] to our task. To be more concrete, we take the Gaussian kernel as the perturbed noise distribution on each pairwise distance, *i.e.*, $q_\sigma(\tilde{\boldsymbol{d}}_1|\boldsymbol{g}_2) = \mathbb{E}_{p_{\text{data}}(\boldsymbol{d}_1|\boldsymbol{g}_2)}[q_\sigma(\tilde{\boldsymbol{d}}_1|\boldsymbol{d}_1)]$, where $\sigma$ is the deviation in Gaussian perturbation. One main advantage of using the Gaussian kernel is that the following gradient of conditional log-likelihood has a closed-form formulation: $\nabla_{\tilde{\boldsymbol{d}}_1}\log q_\sigma(\tilde{\boldsymbol{d}}_1|\boldsymbol{d}_1,\boldsymbol{g}_2) = (\boldsymbol{d}_1 - \tilde{\boldsymbol{d}}_1)/\sigma^2$, and the goal of DSM is to train a score network to match it. This trick was first introduced in [69], and has been widely utilized in the score matching applications [58, 59].

To adapt this into our setting, this is essentially saying that we want to train a "distance network", *i.e.*, $s_\theta(\tilde{\boldsymbol{d}}_1|\boldsymbol{g}_2)$, to match the distance perturbation, or we can say it aims at matching the pseudo-force

with the pairwise distances from another aspect. By taking the Fisher divergence as the discrepancy metric and the trick mentioned above, the estimation $s_\theta(\tilde{\boldsymbol{d}}_1, \boldsymbol{g}_2) \approx \nabla_{\tilde{\boldsymbol{d}}_1} \log q(\tilde{\boldsymbol{d}}_1 | \boldsymbol{d}_1, \boldsymbol{g}_2)$ can be simplified to the following:

$$D_F(q_\sigma(\tilde{\boldsymbol{d}}_1|\boldsymbol{g}_2)||p_\theta(\tilde{\boldsymbol{d}}_1|\boldsymbol{g}_2)) = \frac{1}{2} \mathbb{E}_{p_{\text{data}}(\boldsymbol{d}_1|\boldsymbol{g}_2)} \mathbb{E}_{q_\sigma(\tilde{\boldsymbol{d}}_1|\boldsymbol{d}_1,\boldsymbol{g}_2)} \big[ \| s_\theta(\tilde{\boldsymbol{d}}_1, \boldsymbol{g}_2) - \frac{\boldsymbol{d}_1 - \tilde{\boldsymbol{d}}_1}{\sigma^2} + \|^2 \big] + C. \tag{20}$$

**Final objective.** We adopt the following four model training tricks from [38, 58, 59] to stabilize the score matching training process. (1) We carry out the distance denoising at $L$-level of noises. (2) We add a weighting coefficient $\lambda(\sigma) = \sigma^\beta$ for each noise level, where $\beta$ is the annealing factor. (3) We scale the score network by a factor of $1/\sigma$. (4) We sample the exactly same atoms from the two geometry views with masking ratio $r$. Finally, by considering the two directions and all the above tricks, the objective function becomes the follows:

$$\begin{aligned}
\mathcal{L}_{\text{GeoSSL-DDM}} =& \frac{1}{2L} \sum_{l=1}^{L} \sigma_l^\beta \mathbb{E}_{p_{\text{data}}(\boldsymbol{d}_1|\boldsymbol{g}_2)} \mathbb{E}_{q(\tilde{\boldsymbol{d}}_1|\boldsymbol{d}_1,\boldsymbol{g}_2)} \Big[ \Big\| \frac{s_\theta(\tilde{\boldsymbol{d}}_1, \boldsymbol{g}_2)}{\sigma_l} - \frac{\boldsymbol{d}_1 - \tilde{\boldsymbol{d}}_1}{\sigma_l^2} \Big\|_2^2 \Big] \\
&+ \frac{1}{2L} \sum_{l=1}^{L} \sigma_l^\beta \mathbb{E}_{p_{\text{data}}(\boldsymbol{d}_2|\boldsymbol{g}_1)} \mathbb{E}_{q(\tilde{\boldsymbol{d}}_2|\boldsymbol{d}_2,\boldsymbol{g}_1)} \Big[ \Big\| \frac{s_\theta(\tilde{\boldsymbol{d}}_2, \boldsymbol{g}_1)}{\sigma_l} - \frac{\boldsymbol{d}_2 - \tilde{\boldsymbol{d}}_2}{\sigma_l^2} \Big\|_2^2 \Big].
\end{aligned} \tag{21}$$

## C.4 DISCUSSIONS

Using the energy-based model (EBM) to solve MI maximization can open a novel venue, especially for high-structured data like molecular geometry. To solve EBM, existing methods include noise-contrastive estimation (NCE) [25], score matching (SM) [61], etc. To put this under the MI maximization setting, EBM-NCE is essentially a contrastive learning method, where the goal is to align the positive pairs and contrast the negative pairs simultaneously. While EBM-SM or GeoSSL-DDM, is a generative self-supervised learning (SSL) on distance denoising, and it is especially appealing in the field for geometric data representation learning.

**Further interpretation of pseudo-force.** Score matching can be smoothly adopted to 3D geometric setting. Because scores are defined as gradients of the energy function with respect to the atom positions, it can be thought of a form of pseudo-forces. Following this, GeoSSL-DDM, can be viewed as a pseudo-force matching, which is more natural to the molecular structures. However, further understanding of this requires more domain knowledge in understanding or designing of the energy function. This is beyond the score of this paper, and we would like to leave it for future exploration.

**Multi-view pretraining: complementary information with 2D topological graph.** Recently, there have been certain works [38] proving that 3D geometric information is useful for 2D topology. Here we want to conjecture that the reverse direction is also meaningful: 2D topology can be also useful for 3D representation. This may not seem reasonable from the domain perspective, since 2D topology can be heuristically obtained from the 3D geometry, *i.e.*, all the 2D information is redundant to 3D geometry. However, from the machine learning theory perspective [5, 18], this is still helpful in reducing the sample complexity. From a higher level perspective, we want to explicitly point out that such gap between machine learning and scientific domain has been widely existed, and it would be an interesting direction for further exploration.

# D  EXPERIMENTS

In this section, we would like to discuss the experiment details of our work. The main structure is as follows:

- In Appendix D.1, we introduce the computation resources.
- In Appendices D.2 to D.4, we introduce the downstream datasets.
    - Notice that because the performance of QM9 and MD17 is quite stable after fixing the seed (*e.g.*, 42), we we will not run cross-validation. This also follows the main literature [41, 50, 51].
    - Yet, for LBA & LEP, these two datasets are quite small and are very sensitive to data splitting, so we pick up 5 seeds (12, 22, 32, 42, and 52) and run cross-validation on them.
- In Appendix D.5, we list the key hyperparameters for all the pretraining baselines and GeoSSL-DDM.
- In Appendix D.6, we show the empirical results using SchNet as the backbone model.

## D.1  COMPUTATIONAL RESOURCES

We have around 20 V100 GPU cards for computation at an internal cluster. Each job can be finished within 3-24 hours (each job takes one single GPU card).

## D.2  DATASET: QM9

QM9 [46] is a dataset of 134K molecules consisting of 9 heavy atoms. It includes 12 tasks that are related to the quantum properties. For example, U0 and U298 are the internal energies at 0K at 0K and 298.15K respectively, and U298 and G298 are the other two energies that can be transferred from H298 respectively. The other 8 tasks are quantum mechanics related to the DFT process. We follow [50] in preprocessing the dataset (including unit transformation for each task).

Current work is using different data split (in terms of the splitting size). Originally there are 133,885 molecules in QM9, where 3,054 are filtered out, leading to 130,831 molecules. During the benchmark, we find that the performance on QM9 is very robust to either using (1) 110K for training, 10K for val, 10,831 for test or using (2) 100K for training, 13,083 for val and 17,748 for test. In this paper, we are using option (1).

## D.3  DATASET: MD17

MD17 [10] is a dataset on molecular dynamics simulation. It includes eight tasks, corresponding to eight organic molecules, and each task includes the molecule positions along the potential energy surface (PES), as shown in Figure 1. The goal is to predict the energy-conserving interatomic forces for each atom at each molecule position. We list some basic statistics in Table 6. We follow [41, 51] in preprocessing the dataset (including unit transformation for each task).

Table 6: Some basic statistics on MD17.

| Pretraining | Aspirin ↓ | Benzene ↓ | Ethanol ↓ | Malonaldehyde ↓ | Naphthalene ↓ | Salicylic ↓ | Toluene ↓ | Uracil ↓ |
|---|---|---|---|---|---|---|---|---|
| Train | 1K | 1K | 1K | 1K | 1K | 1K | 1K | 1K |
| Validation | 1K | 1K | 1K | 1K | 1K | 1K | 1K | 1K |
| Test | 209,762 | 47,863 | 553,092 | 991,237 | 324,250 | 318,231 | 440,790 | 131,770 |

## D.4  DATASET: LBA & LEP

Atom3D [66] is a newly published dataset. It gathers several core tasks for 3D molecules, including binding affinity. The binding affinity prediction is to measure the strength of binding interaction between a small molecule to the target protein. Here we will model both the small molecule and large molecule (protein) with their 3D atom coordinates provided.

During the binding process, a cavity in a protein can potentially possess suitable properties for binding a small molecule (ligand), and it is termed a pocking [62]. Because of the large volume of

Table 7: Some basic statistics on LBA & LEP. For LBA, we use split-by-sequence-identity-30: we split protein-ligand complexes such that no protein in the test dataset has more than 30% sequence identity with any protein in the training dataset. For LEP, we split the complex pairs by protein target.

| Pretraining | LBA | LEP |
|---|---|---|
| Train | 3,507 | 304 |
| Validation | 466 | 110 |
| Test | 490 | 104 |
| Split | split-by-identity-30 | split-by-target |

protein, we follow [66] by only taking the binding pocket, where there are no more than 600 atoms for each molecule and protein pair. To be more concrete, we consider two binding affinity tasks. (1) The first task is ligand binding affinity (LBA). It is gathered from [70] and the task is to predict the binding affinity strength between a small molecule and a protein pocket. (2) The second task is ligand efficacy prediction (LEP). We have a molecule bounded to pockets, and the goal is to detect if the same molecule has a higher binding affinity with one pocket compared to the other one. We list some basic statistics in Table 7.

### D.5 HYPERPARAMETER SPECIFICATION

We list all the detailed hyperparameters in this subsection. For all the methods, we use the same optimization strategy, *i.e.*, with learning rate as 5e-4 and cosine annealing learning rate schedule [43]. The other hyperparameters for each pretraining method are listed in Table 8. For the other hyperparameters, we are using the default hyperparameters, as attached in the codes.

Table 8: Hyperparameter specifications.

| Pretraining | Hyperparameter | Value |
|---|---|---|
| Supervised | task | {total energy} |
| Type Prediction | masking ratio | {0.15, 0.3} |
| Distance Prediction | prediction rate | {1} |
| Angle Prediction | prediction rate | {1e-3, 1e-4} |
| RR | perturbed noise $\mu$
perturbed noise $\sigma$
masking ratio $r$ | {0}
{0.3}
{0, 0.3} |
| InfoNCE | perturbed noise $\mu$
perturbed noise $\sigma$
masking ratio $r$ | {0}
{0.3, 1}
{0, 0.3} |
| EBM-NCE | perturbed noise $\mu$
perturbed noise $\sigma$
masking ratio $r$ | {0}
{0.3, 1}
{0, 0.3} |
| GeoSSL-DDM | perturbed noise $\mu$
perturbed noise $\sigma$
masking ratio $r$
$L$
$\sigma_1$
$\sigma_L$
annealing factor $\beta$ | {0}
{0.3}
{0, 0.3}
{30, 50}
{0.01}
{10}
{0.05, 0.2, 2, 5, 10} |

## D.6 SCHNET AS BACKBONE MODEL

We want to highlight that some backbone models (*e.g.*, DimeNet++ and SphereNet) may perform better or on par with the PaiNN, as shown in Table 4. Yet they will be out of GPU memory. Thus, considering all (including the model performance, computation efficiency, and memory cost) together, we adopt PaiNN as the backbone model in the main paper.

In this section, we carry out experiments using SchNet as the backbone model. We follow the same process as in Section 5, *i.e.*, we compare our method with one randomly-initialized and seven pretraining baselines. The results on QM9, MD17, LBA and LEP are in Tables 9 to 11 accordingly. From these three tables, we can observe that in general, GeoSSL-DDM can reach the most optimal results, yielding 21 best performance in 22 downstream tasks, and can reach comparative performance on the remaining task (within top 2 model). This can largely support the effectiveness of our proposed method, GeoSSL-DDM. In addition, we also want to mention that a lot of pretraining tasks show the negative transfer issue. Comparing to the results in Section 5, we conjecture that this is related to the task (both pretraining and downstream tasks) and the backbone model. Yet, this is beyond the scope of our work, and we would like to leave this as a future direction.

Table 9: Downstream results on 12 quantum mechanics prediction tasks from QM9. We take 110K for training, 10K for validation, and 11K for test. The evaluation is mean absolute error, and the best results are in **bold**.

| Pretraining | Alpha ↓ | Gap ↓ | HOMO ↓ | LUMO ↓ | Mu ↓ | Cv ↓ | G298 ↓ | H298 ↓ | R2 ↓ | U298 ↓ | U0 ↓ | Zpve ↓ |
|---|---|---|---|---|---|---|---|---|---|---|---|---|
| – | 0.070 | 50.59 | 32.53 | 26.33 | 0.029 | 0.032 | 14.68 | 14.85 | 0.122 | 14.70 | 14.44 | 1.698 |
| Supervised | 0.070 | 51.34 | 32.62 | 27.61 | 0.030 | 0.032 | 14.08 | 14.09 | 0.141 | 14.13 | 13.25 | 1.727 |
| Type Prediction | 0.084 | 56.07 | 34.55 | 30.65 | 0.040 | 0.034 | 18.79 | 19.39 | 0.201 | 19.29 | 18.86 | 2.001 |
| Distance Prediction | 0.068 | 49.34 | 31.18 | 25.52 | 0.029 | 0.032 | 13.93 | 13.59 | 0.122 | 13.64 | 13.18 | 1.676 |
| Angle Prediction | 0.084 | 57.01 | 37.51 | 30.92 | 0.037 | 0.034 | 15.81 | 15.89 | 0.149 | 16.41 | 15.76 | 1.850 |
| 3D InfoGraph | 0.076 | 53.33 | 33.92 | 28.55 | 0.030 | 0.032 | 15.97 | 16.28 | 0.117 | 16.17 | 15.96 | 1.666 |
| GeoSSL-RR | 0.073 | 52.57 | 34.44 | 28.41 | 0.033 | 0.038 | 15.74 | 16.11 | 0.194 | 15.58 | 14.76 | 1.804 |
| GeoSSL-InfoNCE | 0.075 | 53.00 | 34.29 | 27.03 | 0.029 | 0.033 | 15.67 | 15.53 | 0.125 | 15.79 | 14.94 | 1.675 |
| GeoSSL-EBM-NCE | 0.073 | 52.86 | 33.74 | 28.07 | 0.031 | 0.032 | 14.02 | 13.65 | 0.121 | 13.70 | 13.45 | 1.677 |
| GeoSSL-DDM (ours) | **0.066** | **48.59** | **30.83** | **25.27** | **0.028** | **0.031** | **13.06** | **12.33** | **0.117** | **12.48** | **12.06** | **1.631** |

Table 10: Downstream results on 8 force prediction tasks from MD17. We take 1K for training, 1K for validation, and the number of molecules for test are varied among different tasks, ranging from 48K to 991K. The evaluation is mean absolute error, and the best results are in **bold**.

| Pretraining | Aspirin ↓ | Benzene ↓ | Ethanol ↓ | Malonaldehyde ↓ | Naphthalene ↓ | Salicylic ↓ | Toluene ↓ | Uracil ↓ |
|---|---|---|---|---|---|---|---|---|
| – | 1.196 | 0.404 | 0.542 | 0.879 | 0.534 | 0.786 | 0.562 | 0.730 |
| Supervised | 1.863 | 0.413 | 0.512 | 1.254 | 0.846 | 1.005 | **0.529** | 0.899 |
| Type Prediction | 1.293 | 0.787 | 0.547 | 0.879 | 1.030 | 1.076 | 0.614 | 0.738 |
| Distance Prediction | 1.414 | 0.453 | 0.845 | 1.371 | 0.591 | 0.819 | 0.588 | 0.993 |
| Angle Prediction | 3.030 | 0.450 | 0.485 | 0.845 | 1.112 | 1.214 | 0.791 | 1.016 |
| 3D InfoGraph | 1.545 | 0.448 | 0.640 | 1.080 | 0.827 | 1.096 | 0.735 | 0.760 |
| GeoSSL-RR | 1.878 | 0.450 | 0.690 | 2.255 | 0.960 | 1.382 | 0.784 | 1.188 |
| GeoSSL-InfoNCE | 1.286 | 0.396 | 0.512 | 1.007 | 0.778 | 1.060 | 0.667 | 0.933 |
| GeoSSL-EBM-NCE | 1.271 | 0.400 | 0.570 | 0.972 | 0.605 | 0.862 | 0.576 | 0.790 |
| GeoSSL-DDM (ours) | **1.176** | **0.368** | **0.434** | **0.779** | **0.460** | **0.700** | 0.561 | **0.679** |

Table 11: Downstream results on 2 binding affinity tasks. We select three evaluation metrics for LBA: the root mean squared error (RMSD), the Pearson correlation ($R_p$) and the Spearman correlation ($R_S$). LEP is a binary classification task, and we use the area under the curve for receiver operating characteristics (ROC) and precision-recall (PR) for evaluation. We run cross-validation with 5 seeds, and the best results are in **bold**.

| Pretraining | LBA | | | LEP | |
|---|---|---|---|---|---|
| | RMSD ↓ | $R_P$ ↑ | $R_C$ ↑ | ROC ↑ | PR ↑ |
| – | $1.489 \pm 0.02$ | $0.522 \pm 0.01$ | $0.501 \pm 0.01$ | $0.436 \pm 0.03$ | $0.369 \pm 0.02$ |
| Supervised | $1.477 \pm 0.04$ | $0.528 \pm 0.02$ | $0.503 \pm 0.03$ | $0.462 \pm 0.05$ | $0.392 \pm 0.03$ |
| Type Prediction | $1.483 \pm 0.04$ | $0.498 \pm 0.03$ | $0.481 \pm 0.03$ | $0.570 \pm 0.04$ | $0.509 \pm 0.07$ |
| Distance Prediction | $1.461 \pm 0.06$ | $0.535 \pm 0.04$ | $0.512 \pm 0.04$ | $0.502 \pm 0.06$ | $0.415 \pm 0.05$ |
| Angle Prediction | $1.499 \pm 0.01$ | $0.475 \pm 0.01$ | $0.462 \pm 0.02$ | $0.532 \pm 0.06$ | $0.449 \pm 0.03$ |
| 3D InfoGraph | $1.467 \pm 0.06$ | $0.526 \pm 0.03$ | $0.500 \pm 0.03$ | $0.515 \pm 0.05$ | $0.412 \pm 0.04$ |
| GeoSSL-RR | – | – | – | $0.439 \pm 0.04$ | $0.365 \pm 0.02$ |
| GeoSSL-InfoNCE | $1.528 \pm 0.05$ | $0.483 \pm 0.02$ | $0.464 \pm 0.02$ | $0.588 \pm 0.06$ | $0.523 \pm 0.05$ |
| GeoSSL-EBM-NCE | $1.499 \pm 0.03$ | $0.509 \pm 0.02$ | $0.498 \pm 0.02$ | $0.493 \pm 0.07$ | $0.429 \pm 0.06$ |
| GeoSSL-DDM (ours) | $\mathbf{1.432 \pm 0.02}$ | $\mathbf{0.550 \pm 0.02}$ | $\mathbf{0.529 \pm 0.02}$ | $\mathbf{0.633 \pm 0.03}$ | $\mathbf{0.541 \pm 0.03}$ |

# E  ABLATION STUDIES

## E.1  THE EFFECT OF ANNEALING FACTOR IN GEOSSL-DDM

Among all the hyperparameters (see Table 8) for GeoSSL-DDM, we find that the annealing factor is one of the most sensitive ones. Annealing factor $\beta$ is applied on the weighting coefficient $\lambda(\sigma) = \sigma^\beta$. In this section, we carry out an ablation study to verify this by pretraining GeoSSL-DDM with annealing factors at five different scales.

Table 12: Ablation study on the effect of annealing factor $\beta$ on 12 quantum mechanics prediction tasks from QM9. We take 110K for training, 10K for validation, and 11K for test. The backbone model is PaiNN, and the evaluation is the mean absolute error.

| $\beta$ | Alpha↓ | Gap↓ | HOMO↓ | LUMO↓ | Mu↓ | Cv↓ | G298↓ | H298↓ | r2↓ | U298↓ | U0↓ | Zpve↓ |
|---|---|---|---|---|---|---|---|---|---|---|---|---|
| 0.05 | 0.047 | 40.10 | 23.71 | 19.40 | 0.016 | 0.025 | 7.72 | 7.15 | 0.131 | 7.30 | 7.07 | 1.312 |
| 0.2 | 0.046 | 40.22 | 23.48 | 19.42 | 0.015 | 0.024 | 7.65 | 7.09 | 0.122 | 6.99 | 6.92 | 1.307 |
| 2 | 0.049 | 40.88 | 23.96 | 19.89 | 0.015 | 0.029 | 8.60 | 7.95 | 0.136 | 7.81 | 7.62 | 1.357 |
| 5 | 0.056 | 45.01 | 26.36 | 20.68 | 0.016 | 0.030 | 9.97 | 9.56 | 0.136 | 9.81 | 9.46 | 1.597 |
| 10 | 0.055 | 44.41 | 26.87 | 21.13 | 0.015 | 0.027 | 10.42 | 9.48 | 0.133 | 9.42 | 9.47 | 1.592 |

As can be observed in Table 12, the models are more stable with smaller annealing values (*e.g.*, 0.2 and 0.05). With large annealing values, the model performance can degrade drastically.

## E.2  THE EFFECT ON THE NUMBER OF NOISE LAYERS IN GEOSSL-DDM

Another important hyperparameter listed in Table 8 is the number of noise layers, $L$. Here we conduct an ablation study on it, and the results are shown in Table 13.

Table 13: Ablation study on the effect of the noise layer $L$ on 12 quantum mechanics prediction tasks from QM9. We take 110K for training, 10K for validation, and 11K for test. The backbone model is PaiNN, and the evaluation is the mean absolute error.

| $L$ | Alpha↓ | Gap↓ | HOMO↓ | LUMO↓ | Mu↓ | Cv↓ | G298↓ | H298↓ | r2↓ | U298↓ | U0↓ | Zpve↓ |
|---|---|---|---|---|---|---|---|---|---|---|---|---|
| – (random init) | 0.048 | 44.50 | 26.00 | 21.11 | 0.016 | 0.025 | 8.31 | 7.67 | 0.132 | 7.77 | 7.89 | 1.322 |
| 1 | 0.052 | 42.75 | 25.12 | 20.46 | 0.015 | 0.027 | 9.40 | 9.08 | 0.121 | 8.73 | 8.80 | 1.585 |
| 30 | 0.048 | 40.08 | 23.95 | 19.71 | 0.016 | 0.025 | 8.16 | 7.48 | 0.137 | 7.42 | 7.17 | 1.311 |
| 50 | 0.046 | 40.22 | 23.48 | 19.42 | 0.015 | 0.024 | 7.65 | 7.09 | 0.122 | 6.99 | 6.92 | 1.307 |

In Table 13, we can observe that in general, GeoSSL-DDM can attain better performance with more denoising layers. This is in fact consistent with that in vision applications [61]. Promisingly, even with smaller $L$ (*e.g.*, $L = 1$), GeoSSL-DDM can still achieve a modest improvement to some extent.

# F  STRONG MODEL ROBUSTNESS WITH RANDOM SEEDS

To further illustrate that our proposed GeoSSL-DDM is robust and insensitive to certain random seeds, we further provide the downstream results with more random seeds. We list the key details as follows:

- **Dataset.** We conduct downstream experiments with random seeds on two datasets: QM9 and MD17.
- **Backbone models.** We run two backbone models: PaiNN in Appendix F.1 and SchNet in Appendix F.2.
- **Seeds.** Up till now, for both the main tables (Tables 1, 2, 9 and 10) and ablation studies (in Appendix E), we use a fixed seed 42. In this section, we provide results with two additional seeds 22 and 32.
- **Baselines.** We here compare against the most optimal baselines: random initialization (without any pretraining), distance prediction, representation reconstruction (RR), and EBM-NCE.
- **Reported results.** We report both the mean and standard deviation with seeds 22, 32, and 42 for all the experiments.

## F.1  PaiNN

Here we take the PaiNN as the backbone model. The results on QM9 and MD17 are reported in Tables 14 and 15 respectively. Such empirical results match with the main result in Tables 1 and 2, and they do verify that our proposed GeoSSL-DDM is indeed learning a more robust representation.

Table 14: Downstream results on 12 quantum mechanics prediction tasks from QM9. We take 110K for training, 10K for validation, and 11K for test. The evaluation metric is mean absolute error, and the best results are in **bold**. We report both the mean and standard deviation for seeds 22, 32, and 42.

| Pretraining | Alpha ↓ | Gap ↓ | HOMO↓ | LUMO↓ | Mu ↓ | Cv ↓ | G298 ↓ | H298 ↓ | R2 ↓ | U298 ↓ | U0 ↓ | Zpve ↓ |
|---|---|---|---|---|---|---|---|---|---|---|---|---|
| – | 0.050 ± 0.00 | 44.41 ± 0.75 | 25.81 ± 0.17 | 21.50 ± 0.31 | 0.016 ± 0.00 | **0.025 ± 0.00** | 8.27 ± 0.17 | 7.78 ± 0.24 | 0.134 ± 0.01 | 7.82 ± 0.04 | 7.93 ± 0.23 | 1.310 ± 0.01 |
| Supervised | 0.049 ± 0.00 | 44.27 ± 0.78 | 26.90 ± 0.25 | 21.85 ± 0.09 | 0.017 ± 0.00 | 0.026 ± 0.00 | 8.94 ± 0.11 | 8.54 ± 0.11 | 0.167 ± 0.01 | 8.40 ± 0.13 | 8.25 ± 0.07 | 1.381 ± 0.05 |
| Distance Prediction | 0.062 ± 0.00 | 51.96 ± 4.53 | 28.38 ± 0.80 | 22.63 ± 0.23 | 0.234 ± 0.30 | 0.070 ± 0.05 | 12.39 ± 0.27 | 12.63 ± 0.23 | 0.308 ± 0.23 | 12.28 ± 0.45 | 12.08 ± 0.20 | 1.745 ± 0.07 |
| GeoSSL-RR | 0.047 ± 0.00 | 44.70 ± 0.69 | 25.50 ± 0.06 | 21.35 ± 0.41 | **0.015 ± 0.00** | **0.025 ± 0.00** | 8.57 ± 0.23 | 8.03 ± 0.26 | 0.141 ± 0.01 | 8.21 ± 0.93 | 7.75 ± 0.11 | 1.317 ± 0.03 |
| GeoSSL-InfoNCE | 0.055 ± 0.00 | 45.37 ± 0.20 | 26.83 ± 0.10 | 21.95 ± 0.24 | 0.017 ± 0.00 | 0.044 ± 0.03 | 17.22 ± 11.44 | 17.97 ± 12.47 | 0.514 ± 0.55 | 17.79 ± 12.86 | 17.42 ± 12.59 | 1.902 ± 0.58 |
| GeoSSL-EBM-NCE | 0.049 ± 0.00 | 44.18 ± 0.31 | 26.15 ± 0.17 | 21.77 ± 0.23 | **0.015 ± 0.00** | 0.026 ± 0.00 | 8.79 ± 0.20 | 8.25 ± 0.14 | 0.131 ± 0.00 | 8.21 ± 0.15 | 8.27 ± 0.26 | 1.428 ± 0.02 |
| GeoSSL-DDM (ours) | **0.045 ± 0.00** | **40.29 ± 0.29** | **23.42 ± 0.09** | **19.52 ± 0.13** | **0.015 ± 0.00** | **0.025 ± 0.00** | **7.75 ± 0.16** | **7.17 ± 0.13** | **0.124 ± 0.00** | **7.15 ± 0.15** | **6.98 ± 0.11** | **1.292 ± 0.01** |

Table 15: Downstream results on 8 force prediction tasks from MD17. We take 1K for training, 1K for validation, and the number of molecules for test are varied among different tasks, ranging from 48K to 991K. The evaluation is mean absolute error, and the best results are in **bold**. We report both the mean and standard deviation for seeds 22, 32, and 42.

| Pretraining | Aspirin ↓ | Benzene ↓ | Ethanol ↓ | Malonaldehyde ↓ | Naphthalene ↓ | Salicylic ↓ | Toluene ↓ | Uracil ↓ |
|---|---|---|---|---|---|---|---|---|
| – | 0.559 ± 0.01 | 0.052 ± 0.00 | 0.220 ± 0.01 | 0.338 ± 0.00 | 0.138 ± 0.00 | 0.293 ± 0.00 | 0.156 ± 0.00 | 0.201 ± 0.01 |
| Supervised | 0.507 ± 0.02 | 0.180 ± 0.08 | 0.312 ± 0.03 | 0.480 ± 0.04 | 0.299 ± 0.11 | 0.469 ± 0.07 | 0.238 ± 0.03 | 0.435 ± 0.02 |
| Distance Prediction | 1.701 ± 0.20 | 0.146 ± 0.04 | 0.368 ± 0.07 | 0.757 ± 0.23 | 0.734 ± 0.07 | 1.493 ± 0.35 | 0.340 ± 0.04 | 0.766 ± 0.29 |
| GeoSSL-RR | 0.527 ± 0.04 | 0.053 ± 0.00 | 0.223 ± 0.01 | 0.342 ± 0.02 | 0.136 ± 0.01 | 0.296 ± 0.01 | 0.149 ± 0.01 | 0.190 ± 0.01 |
| GeoSSL-InfoNCE | 0.999 ± 0.11 | 0.108 ± 0.03 | 0.263 ± 0.02 | 0.469 ± 0.06 | 0.415 ± 0.16 | 0.516 ± 0.08 | 0.189 ± 0.01 | 0.506 ± 0.04 |
| GeoSSL-EBM-NCE | 0.724 ± 0.10 | 0.105 ± 0.02 | 0.267 ± 0.02 | 0.479 ± 0.03 | 0.362 ± 0.13 | 0.517 ± 0.13 | 0.241 ± 0.07 | 0.468 ± 0.02 |
| GeoSSL-DDM (ours) | **0.439 ± 0.01** | **0.051 ± 0.00** | **0.170 ± 0.00** | **0.290 ± 0.01** | **0.133 ± 0.01** | **0.267 ± 0.00** | **0.122 ± 0.00** | **0.192 ± 0.01** |

## F.2 SCHNET

Here we take the SchNet as the backbone model. The results on QM9 and MD17 are reported in Tables 16 and 17 respectively. Such empirical results match with the main result in Tables 9 and 10, and they do verify that our proposed GeoSSL-DDM is indeed learning a more robust representation.

Table 16: Downstream results on 12 quantum mechanics prediction tasks from QM9. We take 110K for training, 10K for validation, and 11K for test. The evaluation is mean absolute error, and the best results are in **bold**. We report both the mean and standard deviation for seeds 22, 32, and 42.

| Pretraining | Alpha ↓ | Gap ↓ | HOMO ↓ | LUMO ↓ | Mu ↓ | Cv ↓ | G298 ↓ | H298 ↓ | R2 ↓ | U298 ↓ | U0 ↓ | Zpve ↓ |
|---|---|---|---|---|---|---|---|---|---|---|---|---|
| – | 0.070 ± 0.00 | 50.19 ± 0.54 | 32.35 ± 0.35 | 26.11 ± 0.31 | 0.029 ± 0.00 | 0.032 ± 0.00 | 14.66 ± 0.12 | 14.67 ± 0.25 | 0.129 ± 0.01 | 14.40 ± 0.21 | 14.14 ± 0.22 | 1.699 ± 0.02 |
| Supervised | 0.069 ± 0.00 | 51.07 ± 0.34 | 32.20 ± 0.37 | 27.42 ± 0.17 | 0.030 ± 0.00 | 0.032 ± 0.00 | 14.08 ± 0.11 | 13.92 ± 0.18 | 0.142 ± 0.00 | 13.96 ± 0.14 | 13.41 ± 0.12 | 1.715 ± 0.03 |
| Distance Prediction | 0.067 ± 0.00 | 49.59 ± 0.32 | 31.17 ± 0.04 | 26.08 ± 0.40 | 0.029 ± 0.00 | 0.032 ± 0.00 | 13.81 ± 0.10 | 13.45 ± 0.11 | 0.129 ± 0.01 | 13.49 ± 0.18 | 13.10 ± 0.13 | 1.678 ± 0.02 |
| GeoSSL-RR | 0.078 ± 0.00 | 53.36 ± 0.56 | 34.83 ± 0.47 | 29.84 ± 1.43 | 0.034 ± 0.00 | 0.036 ± 0.00 | 16.84 ± 0.90 | 15.32 ± 0.67 | 0.203 ± 0.01 | 16.43 ± 0.92 | 15.68 ± 0.72 | 1.809 ± 0.01 |
| GeoSSL-InfoNCE | 0.075 ± 0.00 | 53.27 ± 0.20 | 33.80 ± 0.40 | 27.64 ± 0.47 | 0.029 ± 0.00 | 0.033 ± 0.00 | 15.59 ± 0.06 | 15.40 ± 0.09 | 0.125 ± 0.00 | 15.34 ± 0.32 | 15.24 ± 0.22 | 1.670 ± 0.01 |
| GeoSSL-EBM-NCE | 0.072 ± 0.00 | 52.64 ± 0.37 | 33.47 ± 0.24 | 28.01 ± 0.41 | 0.031 ± 0.00 | 0.032 ± 0.00 | 13.67 ± 0.25 | 13.58 ± 0.10 | 0.124 ± 0.00 | 13.52 ± 0.14 | 13.42 ± 0.12 | 1.661 ± 0.01 |
| GeoSSL-DDM (ours) | **0.066 ± 0.00** | **48.78 ± 0.15** | **30.38 ± 0.32** | **25.52 ± 0.23** | **0.028 ± 0.00** | **0.031 ± 0.00** | **12.80 ± 0.19** | **12.36 ± 0.09** | **0.113 ± 0.00** | **12.53 ± 0.04** | **12.12 ± 0.06** | **1.637 ± 0.01** |

Table 17: Downstream results on 8 force prediction tasks from MD17. We take 1K for training, 1K for validation, and the number of molecules for test are varied among different tasks, ranging from 48K to 991K. The evaluation metric is mean absolute error, and the best results are in **bold**. We report the both the mean and standard deviation for seeds 22, 32, and 42.

| Pretraining | Aspirin ↓ | Benzene ↓ | Ethanol ↓ | Malonaldehyde ↓ | Naphthalene ↓ | Salicylic ↓ | Toluene ↓ | Uracil ↓ |
|---|---|---|---|---|---|---|---|---|
| – | 1.418 ± 0.28 | 0.406 ± 0.00 | 0.528 ± 0.01 | 0.908 ± 0.04 | 0.613 ± 0.06 | 0.854 ± 0.08 | 0.575 ± 0.01 | 0.717 ± 0.02 |
| Supervised | 1.714 ± 0.21 | 0.423 ± 0.04 | 0.517 ± 0.01 | 1.127 ± 0.16 | 0.713 ± 0.14 | 1.114 ± 0.08 | 0.578 ± 0.08 | 0.832 ± 0.12 |
| Distance Prediction | 1.756 ± 0.24 | 0.483 ± 0.02 | 0.813 ± 0.02 | 1.458 ± 0.08 | 0.795 ± 0.15 | 1.074 ± 0.19 | 0.691 ± 0.08 | 1.116 ± 0.09 |
| GeoSSL-RR | 2.082 ± 0.20 | 0.563 ± 0.09 | 0.740 ± 0.05 | 1.795 ± 0.35 | 0.910 ± 0.07 | 1.525 ± 0.24 | 0.847 ± 0.14 | 1.159 ± 0.08 |
| GeoSSL-InfoNCE | 1.375 ± 0.07 | 0.432 ± 0.03 | 0.560 ± 0.05 | 1.101 ± 0.12 | 0.797 ± 0.02 | 1.029 ± 0.02 | 0.706 ± 0.03 | 0.934 ± 0.02 |
| GeoSSL-EBM-NCE | **1.297 ± 0.03** | 0.404 ± 0.00 | 0.569 ± 0.00 | 1.005 ± 0.04 | 0.580 ± 0.02 | 0.840 ± 0.02 | 0.581 ± 0.02 | 0.839 ± 0.04 |
| GeoSSL-DDM (ours) | 1.333 ± 0.23 | **0.379 ± 0.01** | **0.466 ± 0.04** | **0.732 ± 0.03** | **0.566 ± 0.13** | **0.824 ± 0.16** | **0.566 ± 0.05** | **0.682 ± 0.07** |

# G    COMPARISON WITH A PARALLEL WORK

We note that there is a parallel work introduced in [76], which also explores the effect of denoising for geometric data pretraining. That work is different from GeoSSL-DDM and we here summarize the main differences as follows:

- The parallel work as presented in [76] is similar to that of denoising score matching (DSM) as introduced in [69], *i.e.*, with only one layer of denoising in score matching. On the contrary, our model has multiple denoising layers, which is much closer to the NCSN [58], where the number of noise layers has been proven to be important to the effectiveness of the denoising score matching models. We here also empirically verify the above analysis. That is, we present the experimental results in Table 13, where $L = 1$ is equivalent to the method in [76]. We can observe that with layer number $L = 1$ (namely the third row of the table), the performance does increase in some cases, which matches with the observation in [76]. Nevertheless, the results in Table 13 clearly indicate that with larger $L$, the model can attain further error reduction and improve model robustness.
- Theoretically, the work in [76] specifically aims at the application task of representation learning in geometric pretraining, through a straightforward adaption of denoising score matching from vision. In contrast, our GeoSSL-DDM approach indeed provides a very general framework that leverages energy-based model (EBM) for mutual information (MI) maximization for geometric data pretraining. As such, GeoSSL-DDM can be easily replaced by other EBM models such as the GFlowNet network [3] to better capture the multi-mode distributions in geometric data during pretraining (please see Section 6 for more discussion).

