# OpenReview forum: "Molecular Geometry Pretraining with SE(3)-Invariant Denoising Distance Matching"
_ICLR.cc/2023/Conference — ICLR 2023 poster_

### Official Review · Reviewer_WmAR · 2022-10-14

**Confidence:** 3
**Correctness:** 3
**Technical Novelty And Significance:** 3
**Empirical Novelty And Significance:** 3
**Recommendation:** 6

**Clarity, Quality, Novelty And Reproducibility:**

The work is novel, albeit it needs to be put in the context of other related studies (see summary of the review).

Additionally, the manuscript could use additional time for rewriting, as it reads very clunkily at times (see last paragraph of section 2, or the energy-based model paragraph on section 3 for examples).

There is no code provided to support the claims or help reproduce the study, as authors claim this will be released in the future.

**Strength And Weaknesses:**

Strengths:
* Strong empirical results
* Novel idea
* Clear methodology
* Evaluation on several relevant biophysical datasets

Weaknesses:
* Hard to read at times
* Does not reference a very related work, while claiming to be the first study to perform pretraining on 3D molecular data.
* Does not provide code to support analyses


**Summary Of The Paper:**

In the presented work, the authors present an approach to perform self supervised learning on tasks where molecular 3D geometry is relevant. They do so by taking advantage of equivariant neural networks in combination with a denoising score matching method on molecular distances.

**Summary Of The Review:**

Overall, the work is original and in my belief deserves publication. However, I think the authors should probably reconsider some of the claims presented in the study, specifically those regarding them being the first to use pretraining strategies on 3D molecular data. Specifically, I believe that claim should be adapted based on the following works:

https://arxiv.org/abs/2206.00133?context=q-bio

The authors should either compare to these approaches or alternatively discuss why their approach is theoretically different, or whether there are computational disadvantages in either of them.

Some other comments/questions:

•	It is not clear how the perturbed geometry g2 is obtained from g1 in Algorithm 1.

•	Do the authors have any suggestions on how to choose the different noise levels \sigma_l in Algorithm 1?

•	It is not clear from the text whether the Molecule3D dataset is a subset of the PubChemQC database.

•	It is also not clear what the authors are referring to when they refer to the so-callsed “supervised pretraining baseline”. In particular I do not see where the pretraining is done here. From the text it appears that this is simply a supervised baseline?

•	It would be fantastic if the authors could provide standard deviations for the numbers reported in Tables 1, 2, as they are reported in 3.

•	In the downstream tasks on binding affinity prediction, is the pretraining done on both the pocket + ligand graph, or only on the ligand? Have the authors experimented with both of these approaches? In this section too, it is not clear to me what the second task (LEP) is. What is an active/inactive conformer of a protein? Are the authors referring to an active/inactive classification tasks for ligands (i.e. as in virtual screening)? Are the same molecules considered in both tasks?

•	Some references do not seem to be present in the main manuscript (e.g. 20)

---

> ### Author Response · Authors · 2022-11-10
> **Thank you for the thoroughful comments (2/2)**
>
>
> **3. The reviewer raises several detailed concerns.**
>
> Thank you for the detailed questions. We are happy to address all of them as below.
>
> - Rewriting of `the last paragraph of section 2 and the EBM paragraph of section 3` We have these paragraphs revised in the latest draft (in blue fonts). Now, they are more concise and precise.
> - `how g2 is obtained from g1`. We revised this in the latest revision (in red fonts).
> - `how to choose noise levels in Algorithm 1`. In the current version, we are taking $L=30/50$ different $\sigma$ from 0.01 to 10, using `linspace` in numpy. In general, we do believe it's worth digging further on what's the effect of different $\sigma$, but the computation cost is too high for now.
> - `Molecule3D` Thank you for pointing that out. We have clarified this in the revision (in red fonts).
> - `Supervised pretraining` This is the supervised pretraining, and the pretraining signals are the energy values. The molecules are not randomly positioned in the 3D space, but actually follow certain physics/chemistry rules and thus flow in the PDE (as illustrated in Figure 1). When molecules approach the local minima, they correspond to a stable position, and certain useful quantum properties (e.g., energies, energy gaps, etc) are available at such position. So the `supervised pretraining` here means that we take energy values at the local minima for pretraining.
> - `More random seeds` Thank you for the comment. This is also raised by Reviewer wxWw. We elaborated on the large computation resources required for more random seeds in the response to wxWw (please see our **comment to Question 3 from Reviwer wxWx for detailed discussions**). Meanwhile, we have also **added the random seeds on MD17 and QM9**. Results are presented in **Sec F in the revision**.
> - `Task specification` (1) We are using the molecule geometries (atom types and atom coordinates) on both the pocket and ligand, following the setting from Atom3D. Using ligand only may be impossible here because of the following comment. (2) LEP task aims for a setting that we have a molecule bounded to two structures (pockets are provided), and the goal is to detect if the same molecule has a higher binding affinity with one pocket comparing to the other one. Namely, both the ligand and pockets are needed in this task. We rephrased this in the revision (in blue fonts) to make it much clearer to the reader.
> - `Reference [20]` We mentioned this paper in appendix C, as it is a good motivation for why adding the noise (as data augmentation) can help get a better geometric representation. We have also added it in Sec 1 now.

---

> ### Author Response · Authors · 2022-11-10
> **Thank you for the thoroughful comments (1/2)**
>
> We appreciate the reviewer for acknowledging our method as novel, our empirical results as strong, and our work as significant for publication. We are also grateful for the reviewer's thoroughful and detailed comments. We  answer the reviewer's questions as follows in detail. We humbly believe that we have successfully addressed all the reviewer's concerns, and hope that the reviewer can increase the evaluation score to reflect the improvement of our paper.
>
> **1. The reviewer concerning our  claim of the first work on 3D SSL.**
>
> We appreciate the reviewer's expertise in the literature. This parallel work [1] (currently is also under peer review) is indeed very related to our proposed GeoSSL method. Now we have added a detailed comparison (both **empirical and theoretical comparison**) between [1] and our GeoSSL in **Sec G in the revised paper**. Below, we highlight the main key points:
>
> - The parallel work in [1] is closer to DSM [2], i.e., **with one layer of denoising for score matching**. While our method has more denoising layers**, and our setting is much closer to the NCSN [3]. In NCSN, **the larger number of noise layers has been proven to be important** to the effectiveness of the denoising score matching models. In addition, we empirically verify the above analysis. That is, we present the **experimental results in Sec F**, where $L=1$ is equivalent to [1]. In that, we do observe that [1] is helpful, yet adding more layers can bring in more benefits.
> - Theoretically, work [1] specifically aims at the application task of representation learning in geometric pretraining, through a straightforward adaption of DSM. In contrast, our GeoSSL approach indeed provides **a very general framework that leverages energy-based model (EBM) for mutual information (MI) maximization for geometric data pretraining**. As such, GeoSSL can be easily replaced by other EBM models such as the GFlowNet network to better capture the multi-mode distributions in geometric data during pretraining (please see Sec 6 for more discussion).
>
> **2. The reviewer also mentions the code publication.**
>
> We appreciate the reviewer evaluating the reproducibility of our work. Now we have **publically shared our source code** on this [anonymous link](https://anonymous.4open.science/r/GeoSSL_rebuttal-00D8). We also want to mention that we keep all the log files and model weights for reproducibility, and they will also be released when the anonymous is not required.
>
> [1] Zaidi, Sheheryar, et al. "Pre-training via Denoising for Molecular Property Prediction." arXiv preprint arXiv:2206.00133 (2022).
>
> [2] Vincent, Pascal. "A connection between score matching and denoising autoencoders." Neural computation 23.7 (2011): 1661-1674.
>
> [3] Song, Yang, and Stefano Ermon. "Generative modeling by estimating gradients of the data distribution." Advances in Neural Information Processing Systems 32 (2019).

---

> ### Comment · Reviewer_WmAR · 2022-11-17
> **Thanks for the thoughtful response**
>
> The authors have successfully addressed most of my comments. I am happy to raise the score to marginally above acceptance.

---

> > ### Author Response · Authors · 2022-11-18
> > **Thank you for raising the score**
> >
> > Thank you very much for raising the score.
> > At the same time, if there is any other question, we would be very glad to further clarify.
> >
> > Regards,
> >
> > Authors of GeoSSL

---

### Official Review · Reviewer_wxWw · 2022-10-20

**Confidence:** 4
**Correctness:** 3
**Technical Novelty And Significance:** 3
**Empirical Novelty And Significance:** 3
**Recommendation:** 5

**Clarity, Quality, Novelty And Reproducibility:**

The paper is written clearly with high quality. The idea is moderately novel. The reproducibility can be improved by providing ablation study on hyperparameters and error bars.

**Strength And Weaknesses:**

Strength: It is a good idea to use denoising in pretraining neural networks in molecular tasks, and the results showed that the idea is valid. The paper is written clearly.

Weakness: The advantage of GeoSSL compared to other pretraining methods seems minor, as seen in Table 1, 2 and 3. I think it is necessary to include two more experiments: (1) ablation study on hyperparameters (2) error bars over multiple random seeds. With these two experiments, it will be more convincing that the advantages are really from pretraining, not hyperparameter-tuning or randomness.

**Summary Of The Paper:**

This paper proposed to use denoising as pretraining for SE(3)-invariant neural network models. The main idea is to add some noise to the coordinates, and use a score model to predict the added noise. The pretrained model are used in several different tasks, including molecule property prediction, force field and binding affinity prediction. The proposed pretraining method was proved better than other pretraining methods in the tasks.

**Summary Of The Review:**

This paper proposed to use denoising in training 3D roto-translation invariant neural networks, and proved its superiority in molecular property prediction, force field and binding affinity tasks. However, the experiments can be more convincing if ablation study and error bars are provided

---

> ### Author Response · Authors · 2022-11-10
> **Ablation studies and random seeds are added (2/2)**
>
>
> **3. The reviewer suggests us to add results with random seeds.**
>
> We appreciate the reviewer's comment on experimenting with random seeds to prove the robustness of GeoSSL. Actually we also found this issue in the molecule geometric modeling community where existing works did not experiemnt with different random seeds. The main reason for that is the incredibly large computation resources required to conduct the cross-validation on these tasks (We detail them as follows). Nevertheless, during the rebuttal, we tried our best to **successfully add the results with random seeds** to abundantently verify the effectiveness of GeoSSL. The **results are presented in Sec F in the revised paper**.
>
> - First we would like to mention that cross-validation results with random seeds are rarely reported on these two tasks (QM9 and MD17). The main bottleneck is the large computation requirements. If the reviewer is interested, you can check the 7 related works (molecule geometric modeling) in Sec A.
> - Meanwhile, in order to verify the effectiveness of our proposed GeoSSL, we have made a comprehensive comparison on 10 pretraining methods (one is randomly-initialized), 4 datasets (22 tasks in total). Further, in order to reduce the biase, we conduct all experiments on two sets of backbone models, PaiNN (Tables 1-3 in Sec 5) and SchNet (Tables 9-11 in Sec D.6).
> - Previously, our experiments adopted the seed from [1] for downstream (QM9 and MD17). [1] is highly reproducible with a fixed and only seed 42. Namely, we are **not tuning the seed to select the best results** for our downstream tasks.
> - In addition, as suggested by the reviewer, we are now trying our best to add QM9 and MD17 results with random seeds. If we want to consider 5 seeds, then:
>   - For the QM9 dataset, each downstream task takes up to 24 hours. Thus, we need 2 (\# backbone models) * 12 (\# tasks) * 10 (\# pretraining methods) * 4 (4 more seeds) * 24 hours = 960 GPU days.
>   - For the MD17 dataset, each downstream task takes around 3 hours to finish. Thus, we need 2 (\# backbone models) * 8 (\# tasks) * 10 (\# pretraining methods) * 4 (4 more seeds) * 3 hours = 80 GPU days to finish.
> - As you can see, the computational cost is huge, and it is impossible to finish them during the rebuttal. Just to mention that it took us over a month to finish running all the downstream tasks for the main results in Tables 1-3 and 9-11.
> - Now in Sec F, we  **report the random-seed (22, 32, and 42) results** on QM9 and MD17, using both optimal baselines and GeoSSL. An example on QM9 is below: (backbone model is PaiNN)
>
> | pretraining method | Alpha | Gap | Homo | Lumo | Mu | Cv | G298 | H298 | r2 | U298 | U0 | Zpve |
> | --- | --- | --- | --- | --- | --- | --- | --- | --- | --- | --- | --- | --- |
> -- | 0.050 $\pm$ 0.00 | 44.41 $\pm$ 0.75 | 25.81 $\pm$ 0.17 | 21.50 $\pm$ 0.31 | 0.016 $\pm$ 0.00 | 0.025 $\pm$ 0.00 | 8.27 $\pm$ 0.17 | 7.78 $\pm$ 0.24 | 0.134 $\pm$ 0.01 | 7.82 $\pm$ 0.04 | 7.93 $\pm$ 0.23 | 1.310 $\pm$ 0.01
> Distance Prediction | 0.062 $\pm$ 0.00 | 51.96 $\pm$ 4.53 | 28.38 $\pm$ 0.80 | 22.63 $\pm$ 0.23 | 0.234 $\pm$ 0.30 | 0.070 $\pm$ 0.05 | 12.39 $\pm$ 0.27 | 12.63 $\pm$ 0.23 | 0.308 $\pm$ 0.23 | 12.28 $\pm$ 0.45 | 12.08 $\pm$ 0.20 | 1.745 $\pm$ 0.07
> RR | 0.047 $\pm$ 0.00 | 44.70 $\pm$ 0.69 | 25.50 $\pm$ 0.06 | 21.35 $\pm$ 0.41 | 0.015 $\pm$ 0.00 | 0.025 $\pm$ 0.00 | 8.57 $\pm$ 0.23 | 8.03 $\pm$ 0.26 | 0.141 $\pm$ 0.01 | 8.21 $\pm$ 0.93 | 7.75 $\pm$ 0.11 | 1.317 $\pm$ 0.03
> EBM-NCE | 0.049 $\pm$ 0.00 | 44.18 $\pm$ 0.31 | 26.15 $\pm$ 0.17 | 21.77 $\pm$ 0.23 | 0.015 $\pm$ 0.00 | 0.026 $\pm$ 0.00 | 8.79 $\pm$ 0.20 | 8.25 $\pm$ 0.14 | 0.131 $\pm$ 0.00 | 8.21 $\pm$ 0.15 | 8.27 $\pm$ 0.26 | 1.428 $\pm$ 0.02
> GeoSSL (ours) | 0.045 $\pm$ 0.00 | 40.29 $\pm$ 0.29 | 23.42 $\pm$ 0.09 | 19.52 $\pm$ 0.13 | 0.015 $\pm$ 0.00 | 0.025 $\pm$ 0.00 | 7.75 $\pm$ 0.16 | 7.17 $\pm$ 0.13 | 0.124 $\pm$ 0.00 | 7.15 $\pm$ 0.15 | 6.98 $\pm$ 0.11 | 1.292 $\pm$ 0.01
>
> ~~PS. We are still running experiments on SchNet, and shall keep updating the random seed results in the revised paper during rebuttal.~~
>
> [1] Liu, Yi, et al. "Spherical message passing for 3d molecular graphs." International Conference on Learning Representations. 2021.

---

> > ### Author Response · Authors · 2022-11-16
> > **Updates on random seeds experiments**
> >
> > Hi there,
> >
> > We have updated the results with random results in Sec F in the latest revision. We added two more baselines and one more backbone model (SchNet) on two datasets. All the results can support the robustness of GeoSSL.
> >
> > We list the results on QM9 below.
> >
> > - **Random seeds on QM9 and PaiNN (two more baselines than the previous response).**
> >
> > | pretraining method | Alpha | Gap | Homo | Lumo | Mu | Cv | G298 | H298 | r2 | U298 | U0 | Zpve |
> > | --- | --- | --- | --- | --- | --- | --- | --- | --- | --- | --- | --- | --- |
> > -- | 0.050 $\pm$ 0.00 | 44.41 $\pm$ 0.75 | 25.81 $\pm$ 0.17 | 21.50 $\pm$ 0.31 | 0.016 $\pm$ 0.00 | 0.025 $\pm$ 0.00 | 8.27 $\pm$ 0.17 | 7.78 $\pm$ 0.24 | 0.134 $\pm$ 0.01 | 7.82 $\pm$ 0.04 | 7.93 $\pm$ 0.23 | 1.310 $\pm$ 0.01
> > Supervised | 0.049 $\pm$ 0.00 | 44.27 $\pm$ 0.78 | 26.90 $\pm$ 0.25 | 21.85 $\pm$ 0.09 | 0.017 $\pm$ 0.00 | 0.026 $\pm$ 0.00 | 8.94 $\pm$ 0.11 | 8.54 $\pm$ 0.11 | 0.167 $\pm$ 0.01 | 8.40 $\pm$ 0.13 | 8.25 $\pm$ 0.07 | 1.381 $\pm$ 0.05
> > Distance Prediction | 0.062 $\pm$ 0.00 | 51.96 $\pm$ 4.53 | 28.38 $\pm$ 0.80 | 22.63 $\pm$ 0.23 | 0.234 $\pm$ 0.30 | 0.070 $\pm$ 0.05 | 12.39 $\pm$ 0.27 | 12.63 $\pm$ 0.23 | 0.308 $\pm$ 0.23 | 12.28 $\pm$ 0.45 | 12.08 $\pm$ 0.20 | 1.745 $\pm$ 0.07
> > RR | 0.047 $\pm$ 0.00 | 44.70 $\pm$ 0.69 | 25.50 $\pm$ 0.06 | 21.35 $\pm$ 0.41 | 0.015 $\pm$ 0.00 | 0.025 $\pm$ 0.00 | 8.57 $\pm$ 0.23 | 8.03 $\pm$ 0.26 | 0.141 $\pm$ 0.01 | 8.21 $\pm$ 0.93 | 7.75 $\pm$ 0.11 | 1.317 $\pm$ 0.03
> > InfoNCE | 0.055 $\pm$ 0.00 | 45.37 $\pm$ 0.20 | 26.83 $\pm$ 0.10 | 21.95 $\pm$ 0.24 | 0.017 $\pm$ 0.00 | 0.044 $\pm$ 0.03 | 17.22 $\pm$ 11.44 | 17.97 $\pm$ 12.47 | 0.514 $\pm$ 0.55 | 17.79 $\pm$ 12.86 | 17.42 $\pm$ 12.59 | 1.902 $\pm$ 0.58
> > EBM-NCE | 0.049 $\pm$ 0.00 | 44.18 $\pm$ 0.31 | 26.15 $\pm$ 0.17 | 21.77 $\pm$ 0.23 | 0.015 $\pm$ 0.00 | 0.026 $\pm$ 0.00 | 8.79 $\pm$ 0.20 | 8.25 $\pm$ 0.14 | 0.131 $\pm$ 0.00 | 8.21 $\pm$ 0.15 | 8.27 $\pm$ 0.26 | 1.428 $\pm$ 0.02
> > GeoSSL (ours) | 0.045 $\pm$ 0.00 | 40.29 $\pm$ 0.29 | 23.42 $\pm$ 0.09 | 19.52 $\pm$ 0.13 | 0.015 $\pm$ 0.00 | 0.025 $\pm$ 0.00 | 7.75 $\pm$ 0.16 | 7.17 $\pm$ 0.13 | 0.124 $\pm$ 0.00 | 7.15 $\pm$ 0.15 | 6.98 $\pm$ 0.11 | 1.292 $\pm$ 0.01
> >
> >
> > - **Random seeds on QM9 and SchNet.**
> >
> > | pretraining method | Alpha | Gap | Homo | Lumo | Mu | Cv | G298 | H298 | r2 | U298 | U0 | Zpve |
> > | --- | --- | --- | --- | --- | --- | --- | --- | --- | --- | --- | --- | --- |
> > -- | 0.070 $\pm$ 0.00 | 50.19 $\pm$ 0.54 | 32.35 $\pm$ 0.35 | 26.11 $\pm$ 0.31 | 0.029 $\pm$ 0.00 | 0.032 $\pm$ 0.00 | 14.66 $\pm$ 0.12 | 14.67 $\pm$ 0.25 | 0.129 $\pm$ 0.01 | 14.40 $\pm$ 0.21 | 14.14 $\pm$ 0.22 | 1.699 $\pm$ 0.02
> > Supervised | 0.069 $\pm$ 0.00 | 51.07 $\pm$ 0.34 | 32.20 $\pm$ 0.37 | 27.42 $\pm$ 0.17 | 0.030 $\pm$ 0.00 | 0.032 $\pm$ 0.00 | 14.08 $\pm$ 0.11 | 13.92 $\pm$ 0.18 | 0.142 $\pm$ 0.00 | 13.96 $\pm$ 0.14 | 13.41 $\pm$ 0.12 | 1.715 $\pm$ 0.03
> > Distance Prediction | 0.067 $\pm$ 0.00 | 49.59 $\pm$ 0.32 | 31.17 $\pm$ 0.04 | 26.08 $\pm$ 0.40 | 0.029 $\pm$ 0.00 | 0.032 $\pm$ 0.00 | 13.81 $\pm$ 0.10 | 13.45 $\pm$ 0.11 | 0.129 $\pm$ 0.01 | 13.49 $\pm$ 0.18 | 13.10 $\pm$ 0.13 | 1.678 $\pm$ 0.02
> > RR | 0.078 $\pm$ 0.00 | 53.36 $\pm$ 0.56 | 34.83 $\pm$ 0.47 | 29.84 $\pm$ 1.43 | 0.034 $\pm$ 0.00 | 0.036 $\pm$ 0.00 | 16.84 $\pm$ 0.90 | 15.32 $\pm$ 0.67 | 0.203 $\pm$ 0.01 | 16.43 $\pm$ 0.92 | 15.68 $\pm$ 0.72 | 1.809 $\pm$ 0.01
> > InfoNCE | 0.075 $\pm$ 0.00 | 53.27 $\pm$ 0.20 | 33.80 $\pm$ 0.40 | 27.64 $\pm$ 0.47 | 0.029 $\pm$ 0.00 | 0.033 $\pm$ 0.00 | 15.59 $\pm$ 0.06 | 15.40 $\pm$ 0.09 | 0.125 $\pm$ 0.00 | 15.34 $\pm$ 0.32 | 15.24 $\pm$ 0.22 | 1.670 $\pm$ 0.01
> > EBM-NCE | 0.072 $\pm$ 0.00 | 52.64 $\pm$ 0.37 | 33.47 $\pm$ 0.24 | 28.01 $\pm$ 0.41 | 0.031 $\pm$ 0.00 | 0.032 $\pm$ 0.00 | 13.67 $\pm$ 0.25 | 13.58 $\pm$ 0.10 | 0.124 $\pm$ 0.00 | 13.52 $\pm$ 0.14 | 13.42 $\pm$ 0.12 | 1.661 $\pm$ 0.01
> > GeoSSL (ours) | 0.066 $\pm$ 0.00 | 48.78 $\pm$ 0.15 | 30.38 $\pm$ 0.32 | 25.52 $\pm$ 0.23 | 0.028 $\pm$ 0.00 | 0.031 $\pm$ 0.00 | 12.80 $\pm$ 0.19 | 12.36 $\pm$ 0.09 | 0.113 $\pm$ 0.00 | 12.53 $\pm$ 0.04 | 12.12 $\pm$ 0.06 | 1.637 $\pm$ 0.01

---

> > > ### Author Response · Authors · 2022-11-24
> > > **Have we addressed your concerns?**
> > >
> > > Dear reviewer,
> > >
> > > Thank you for your constructive comments. As suggested, we have added the ablation studies for hyperparameter tuning and random-seed  (for the two backbone models) to the revision. The newly added empirical results clearly indicate the robustness of our GeoSSL method.
> > >
> > > We wonder if our rebuttal has addressed your concerns. Also, we are happy to answer any further question if you have.

---

> ### Author Response · Authors · 2022-11-10
> **Ablation studies and random seeds are added (1/2)**
>
>
> We appreciate the reviewer for acknowledging this work as novel, the results as valid, and the paper is clearly written. The reviewer's main concerns are about the robustness of GeoSSL. Thus, we have added the request ablation studies and hyperparameter variations. We hope these results are reasonable and satisfactory enough to address the reviewer’s concerns.
>
> **1. The reviewer notes that performance improvements of GeoSSL is minor comparing to other pretraining baselines.**
>
> - First, we want to highlight that for the main results in Tables 1-3 and Tables 9-11, the pretraining comparison baselines are in general pretty bad. As can been seen, there are severe **negative transfer issue** for those comparison baselines. In contrast, our method does not have such problem.
> - Second, our method GeoSSL, in this case, can obtain **much better performance and almost consist improvmenets** over the baseline models. Such improvement and robuseness of our method have been further confirmed by the newly added random-seeded experiments as will be discussed next.
>
> **2. The reviewer asks for more ablation studies on hyperparameters.**
>
> Thank you for the comments. We listed all the hyper-parameters in Table 8, and here we add two main hyperparameters for ablation studies. The high-level discussions are below and more details can be found in **Sec E in the revised paper**.
>
> - The first ablation study is on the **annealing factor $\beta$**, where we find that roughly the smaller annealing factors (0.05 and 0.2) can show more robust results than larger ones ($\ge 2$). We show the effect of $\beta$ in GeoSSL as below (backbone model: PaiNN; dataset: QM9; seed: 42. More details are in **Sec E1 in the revised paper**):
>
> | $\beta$ | Alpha | Gap | Homo | Lumo | Mu | Cv | G298 | H298 | r2 | U298 | U0 | Zpve |
> | --- | --- | --- | --- | --- | --- | --- | --- | --- | --- | --- | --- | --- |
> 0.05 | 0.047 | 40.10 | 23.71 | 19.40 | 0.016 | 0.025 | 7.72 | 7.15 | 0.131 | 7.30 | 7.07 | 1.312
> 0.2 | 0.046 | 40.22 | 23.48 | 19.42 | 0.015 | 0.024 | 7.65 | 7.09 | 0.122 | 6.99 | 6.92 | 1.307
> 2 | 0.049 | 40.88 | 23.96 | 19.89 | 0.015 | 0.029 | 8.60 | 7.95 | 0.136 | 7.81 | 7.62 | 1.357
> 5 | 0.056 | 45.01 | 26.36 | 20.68 | 0.016 | 0.030 | 9.97 | 9.56 | 0.136 | 9.81 | 9.46 | 1.597
> 10 | 0.055 | 44.41 | 26.87 | 21.13 | 0.015 | 0.027 | 10.42 | 9.48 | 0.133 | 9.42 | 9.47 | 1.592
>
> - The second ablation study is on **the number of denoising layers $L$**. We observe that (1) adding more layers can in general lead to a more robust performance, and (2) adding smaller number of layers, though not as good as larger $L$, yet it can still shows a modest performance improvement comparing to the baselines. We show the effect of $L$ on GeoSSL as below (backbone model: PaiNN; dataset: QM9; seed: 42. More details are in **Sec E2 in the revised paper**):
>
> | $L$ | Alpha | Gap | Homo | Lumo | Mu | Cv | G298 | H298 | r2 | U298 | U0 | Zpve |
> | --- | --- | --- | --- | --- | --- | --- | --- | --- | --- | --- | --- | --- |
> |1 | 0.052 | 42.75 | 25.12 | 20.46 | 0.015 | 0.027 | 9.40 | 9.08 | 0.121 | 8.73 | 8.80 | 1.585|
> |30 | 0.048 | 40.08 | 23.95 | 19.71 | 0.016 | 0.025 | 8.16 | 7.48 | 0.137 | 7.42 | 7.17 | 1.311|
> |50 | 0.046 | 40.22 | 23.48 | 19.42 | 0.015 | 0.024 | 7.65 | 7.09 | 0.122 | 6.99 | 6.92 | 1.307

---

### Official Review · Reviewer_a9bJ · 2022-11-03

**Confidence:** 3
**Correctness:** 3
**Technical Novelty And Significance:** 3
**Empirical Novelty And Significance:** 3
**Recommendation:** 6

**Clarity, Quality, Novelty And Reproducibility:**

The description of the method is clear and easy to follow. However, the code is missing. It would be good if the authors can include the computing resources for each experiment.

**Strength And Weaknesses:**


Strengths
1. The derivation from mutual information maximization to denoising score matching is novel.
2. The experiments are convincing.

Weakness
1. In Sec 4.3.1 Eq.(5), the definition of $d$ and $r$ are missing
2. The calculation of $d$ is pariwise distances of all points (or atoms). If the number of points is very large, it could be very slow (quadratic complexity)?
3. It seems that the method can also be used with general point cloud data (no edges between vertices), can the authors elaborate more about this? The complexity seems to be an important problem if we want to apply the method to large point clouds.
4. The main contribution is the denoising distance matching. Is it appropriate to put "SE(3)-invariant" in the title?
5. Why is this method superior to a vanilla denoising autoencoder (the simplest method we can come up with when dealing with SSL)? More discussions are needed
6. How is the memory consumption given different sizes of input graphs?
7. If I am understanding this correctly, the row marked with a dash (-) means not using any pretraining method (Table 1-3). It is just equivalent to training from scratch.
8. In D.1, how many v100 cards do you use for a single experiment?
9. Can the authors discuss a bit more about the limitations of the proposed method?

**Summary Of The Paper:**

This paper proposes a self-supervised method for 3d molecular data. The authors started with maximizing mutual information, and obtained an objective loss (denoising score matching). The idea is supported by several 3d geometric pretraining experiments.

**Summary Of The Review:**

The paper proposed an interesting SSL method. But I am concerned about the computing resources needed due to the quadratic complexity.

---

> ### Author Response · Authors · 2022-11-10
> **Thank you for the comments**
>
> We appreciate the reviewer for acknowleding our method as novel and experiment results as convincing. We are also grateful to the reviewer's thoroughful and detailed comments. We have addressed all the 10 weaknesses listed by the reviewer as below, and humbly hope that the reviewer can consider increasing the evaluation score.
>
> 1. Thank you for pointing out the missing definitions. We have added them in the revised version (red fonts).
> 2. We are not experts on point clouds, but this seems correct. The time complexity is $O(N^2)$, where $N$ is the number of nodes. For molecule data, which is the aim of this paper, the quadratic complexity is not an issue for the following reasons. First, unlike point clouds, the atomic nodes in a molecule is typically very small (e.g., the average **total number of atoms (nodes) per molecule in our training data is 29.11**). Second, if needed, we can always preprocess the dataset on  CPU, so empirically it won't take too much extra time as long as the batch of data can fit into the CPU and GPU memory.
> 4. Yes, the reviewer understands this correctly. Our proposed method can be applied to the points clouds naturally (also in Sec 6), yet the time complexity might be the bottleneck for large point clouds. Point clouds are typically much larger than small molecules we are aiming at in this paper.
> 5. Thank you for the insighful comment. We highlight the SE(3)-invariant in the first place because of the following reasons:
>     - First, our GeoSSL has **two key components**: the molecule geometric modeling (backbone model like PaiNN) and the score network (for the diffusion process like Eq 9 in the paper).
>     - Second, **both** the molecule model and score network **are expected to be SE(3)-equivariant**. Notice that SE(3)-invariance is a special case of SE(3)-equivariance.
>     - Finally, in our work, we first propose a very general framework for geometric data pretraining, the coordinate denoising objective (Eq 3). Then we transform it into a pairwise-distance-denoising problem, which satisfies SE(3)-invariant on the type-0 features. By using SE(3)-invariant, we wanted to **distinguish our method from the SE(3)-equivariant models**, which operate on higher-order features like coordinates or other type-1 features.
> 6. We would like to confirm with the reviewer on what the `denoising autoencoder` method is referred to here.
>     - If it means `coordinate denoising autoencoder`, then in the paper (Sec 4.3), we already discussed that. That is,  directly denoising the coordinates needs the score network to satisfy the SE(3)-equivariant constraint. This is nontrivial especially when the geometric model is SE(3)-invariant like SchNet. Meanwhile, after the decomposition in Eq 5, we can turn it into a SE(3)-invariant model, which can is more trivial.
>     - If the reviewer is referring to the `atom-type denoising autoencoder`, then we have already included this as a baseline in the main results (e.g., `Type Prediction` in Tables 1-3 and Tables 9-11).
> 7. For the memory cost, we use V100 card with 32G CUDA memory, and it can fit with batch-size=128, where the backbone models can be PaiNN or SchNet. The reviewer is correct that the memory cost is a big issue, and we failed to fit them into the larger models, like GemNet.
> 8. Yes, the reviewer understands this correctly.
> 9. For each single experiment, we use only one V100 card.
> 10. One limitation can be the SE(3)-invariant diffusion model (or score network). A lot of works have proven the effectiveness of SE(3)-equivariant network for the geometric modeling [1] (the backbone model in GeoSSL), but none of them has been exploring the effectiveness of the **representation** power of SE(3)-equivariant diffusion model. As a starting work in this research line, we use the SE(3)-invariant model as the first step. In the future, we would like to **explore the SE(3)-equivariant** diffusion model for representation learning.
> 11. We have **publically shared our source code** at this [anonymous link](https://anonymous.4open.science/r/GeoSSL_rebuttal-00D8). We highlight this in the Reproducibility Statement section **in the revised paper (red fonts)**.
>
> [1] Satorras, Vıctor Garcia, Emiel Hoogeboom, and Max Welling. "E(n)equivariant graph neural networks." International conference on machine learning. PMLR, 2021.

---

> > ### Comment · Reviewer_a9bJ · 2022-11-24
> > **Response**
> >
> > Thanks for the clarification. I still lean towards accepting the paper. This paper gives a new framework for molecular self supervised learning. The method is to denoising pairwise distances, which is different from coordinate denoising and attribute denoising. However, it would be good if the authors can emphasize the differences somewhere in the paper (e.g., a figure or a table). There are not many explanatory figures in this paper. It may cause confusion.

---

> > > ### Author Response · Authors · 2022-11-24
> > > **Difference from attribute denoising and coordinate denoising**
> > >
> > >
> > > Thank you for championing our paper. We are happy to provide the detailed comparison below, and will add them in the future version.
> > >
> > > | Method | Figure | Comparison |
> > > | --- | --- | --- |
> > > | Attribute  Denoising | -- | Masking the atomic attributes, and then applying the denoising process to recover them for pretraining. Nevertheless, for geometric data, coordinate information plays a vital role in representation learning, thus coordinate denoising is typically preferred.|
> > > | Coordinate Denoising | **Figure 3** | Adding noise to the atom coordinates, and then performing denoising. For such denoising, we will need an equivariant denoising process. More importantly, directly denoising such noisy coordinates is very challenging because one may need to effectively constrain the pairwise atomic distances while changing the atomic coordinates.|
> > > | Distance Denoising (ours) | **Figure 2/4** | We decompose the above coordinate denoising problem into the distance denoising problem via a novel SE(3)-invariant denoising strategy. By leveraging the  SE(3)-invariant score matching method, we successfully transform the coordinate denoising desire to the denoising of pairwise atomic distances, which then can be effectively computed.|
> > >
> > >
> > > We hope that we have addressed all your questions. If there is any other confusion, we would be very glad to further clarify.

---

### Author Response · Authors · 2022-11-16
**Response Summary**

Dear reviewers and ACs,

We sincerely thank you for your time and effort on reviewing our paper.

We are glad for your positive comments, including **the method is novel** (Reviewers a9bJ, wxWw, and WmAR), the **experiment results are convincing** (Reviewers a9bJ and WmAR), the **paper is clearly written** (Reviewer a9bJ and wxWw), and **the paper deserves publication** (Reviewer WmAR). At the same time, we have carefully revised and improved the manuscript according to your insightful suggestions (revisions presented in red and blue fonts, for the ease of the reviewers). The main updates are listed as follows:
- We have **publically shared our source code** via [this anonymous link](https://anonymous.4open.science/r/GeoSSL_rebuttal-00D8), as suggested by **Reviewers a9bj and WmAR**.
- We  added **ablation studies** on the two key hyper-parameters of our method GeoSSL in Sec E, as requested by **Reviewer wxWw**.
- We added **experimental results with random seeds** in Sec F, as recommended by **Reviewers wxWw and WmAR**.
  - In the rebuttal, we have demonstrated the computation resources required for these tasks (up to 1K GPU days). Meanwhile, we have tried our best to cover 7 methods (6 baselines and GeoSSL) on 2 tasks (QM9 and MD17) and 2 backbone models (PaiNN and SchNet) with 3 random seeds. The empirical results can support the robustness of our proposed GeoSSL.
- We  cited the parallel work in Sec 1 as requested by **Reviewer WmAR**. Additionaly, we  added the methodological and experimental comparison between this work and our proposed GeoSSL in Sec G.
- We revised all the descriptions for better clarity as suggested by **Reviewer WmAR**. We have also  made an effort to further polish the presentation of the paper.

We humbly believe that we have addressed all your comments, and sincerely hope that you could consider increasing your evaluation scores.

---

### Decision · Program_Chairs · 2023-01-20

**Decision:**

Accept: poster

**Justification For Why Not Higher Score:**

Given the borderline reviews, I cannot advocate for a stronger acceptance (spotlight/oral).

**Justification For Why Not Lower Score:**

On the strength of the convincing performance of the method, paired with its novelty, I vote for acceptance.

**Metareview: Summary, Strengths And Weaknesses:**

The paper proposes a self-supervised representation learning method for small molecules based on denoising the pair-wise distances. The learned representations are shown to improve predictions on downstream tasks. The method was deemed to be novel by the reviewers, and the evaluations convincing. However this was still a borderline paper, mainly due to concerns that the technique is an obvious application of SSL techniques already applied to other problems in ML.

On the strength of the convincing performance of the method, paired with its novelty, I vote for acceptance.

**Note From Pc:**

if the above contains the word "oral" or "spotlight" please see: "oral" presentation means -> notable-top-5% and "spotlight" means -> notable-top-25%. As stated in our emails, we are disassociating presentation type from AC recommendations

**Summary Of Ac-Reviewer Meeting:**

[Note: We were unable convene all three reviewers at the same time due to technical issues. One joined the video conference, and another sent comments by email.]

The reviewers were in agreement that the paper proposed a novel method, that the method was evaluated appropriately, and that the results constitute an advance in the domain of application. The main difference of opinion stemmed from whether the method was not sufficiently novel, and perhaps a relatively straightforward translation of an SSL method already applied to other domains in ML.

Ultimately, since all methods will not be successful in all domains, and since SSL methods are still very much under active development, it will be of interest to the community at large to understand which methods work for what problems, and it will be of great interest to practitioners of this community to learn what techniques work well for this problem. So I recommend acceptance.